# Prodrug-conjugated tumor-seeking commensals for targeted cancer therapy

Haosheng Shen [1,2,3], Changyu Zhang[4,5], Shengjie Li[1,6], Yuanmei Liang[1,2,3,7], Li Ting Lee [1,2,3], Nikhil Aggarwal[1,2,3], Kwok Soon Wun[1,2,3,7], Jing Liu[8], Saravanan Prabhu Nadarajan[1,2,7], Cheng Weng [5], Hua Ling [1,2,7,11], Joshua K. Tay [1,2,9], De Yun Wang[8], Shao Q. Yao [5], In Young Hwang[1,2,12] ✉, Yung Seng Lee [1,2,10] & Matthew Wook Chang [1,2,3,7] ✉

Prodrugs have been explored as an alternative to conventional chemotherapy; however, their target specificity remains limited. The tumor microenvironment harbors a range of microorganisms that potentially serve as tumor-targeting vectors for delivering prodrugs. In this study, we harness bacteria-cancer interactions native to the tumor microbiome to achieve high target specificity for prodrug delivery. We identify an oral commensal strain of *Lactobacillus plantarum* with an intrinsic cancer-binding mechanism and engineer the strain to enable the surface loading of anticancer prodrugs, with nasopharyngeal carcinoma (NPC) as a model cancer. The engineered commensals show specific binding to NPC via OppA-mediated recognition of surface heparan sulfate, and the loaded prodrugs are activated by tumor-associated biosignals to release SN-38, a chemotherapy compound, near NPC. In vitro experiments demonstrate that the prodrug-loaded microbes significantly increase the potency of SN-38 against NPC cell lines, up to 10-fold. In a mouse xenograft model, intravenous injection of the engineered *L. plantarum* leads to bacterial colonization in NPC tumors and a 67% inhibition in tumor growth, enhancing the efficacy of SN-38 by 54%.

Conventional chemotherapy is the primary mode of cancer treatment, but it is frequently linked to poor bioavailability, severe systemic toxicity, and low patient tolerance[1]. To overcome these challenges, prodrugs have been investigated as an alternative approach for targeted cancer therapy[2,3]. Prodrugs are molecules with little pharmacological activity that permit conversion to active native drugs in vivo by chemical or enzymatic reactions. Higher selectivity of prodrugs can be achieved by leveraging physiological conditions unique to the tumor microenvironment (TME), such as hypoxia[4,5], acidosis[6,7], high oxidative stress[8,9] and elevated glutathione (GSH) level[10]. In this case, cytotoxic drugs are modified by chemical groups that are responsive to TME cues and are released when they reach tumors in a site-specific

[1]NUS Synthetic Biology for Clinical and Technological Innovation (SynCTI), National University of Singapore, Singapore, Singapore. [2]Synthetic Biology Translational Research Programme, Yong Loo Lin School of Medicine, National University of Singapore, Singapore, Singapore. [3]National Centre for Engineering Biology (NCEB), Singapore, Singapore. [4]Ningbo Institute of Dalian University of Technology, Ningbo, China. [5]Department of Chemistry, National University of Singapore, Singapore, Singapore. [6]Institute of Translational Medicine, Jiangxi Medical College, Nanchang University, Nanchang, China. [7]Department of Biochemistry, Yong Loo Lin School of Medicine, National University of Singapore, Singapore, Singapore. [8]Department of Otolaryngology, Infectious Diseases Translational Research Program, Yong Loo Lin School of Medicine, National University of Singapore, Singapore, Singapore. [9]Department of Otolaryngology-Head and Neck Surgery, National University of Singapore, Singapore, Singapore. [10]Department of Paediatrics, Yong Loo Lin School of Medicine, National University of Singapore, Singapore, Singapore. [11]Present address: Wilmar International Limited, Singapore, Singapore. [12]Present address: Food, Chemical and Biotechnology, Singapore Institute of Technology, Singapore, Singapore. ✉e-mail: iyhwang@nus.edu.sg; bchcmw@nus.edu.sg

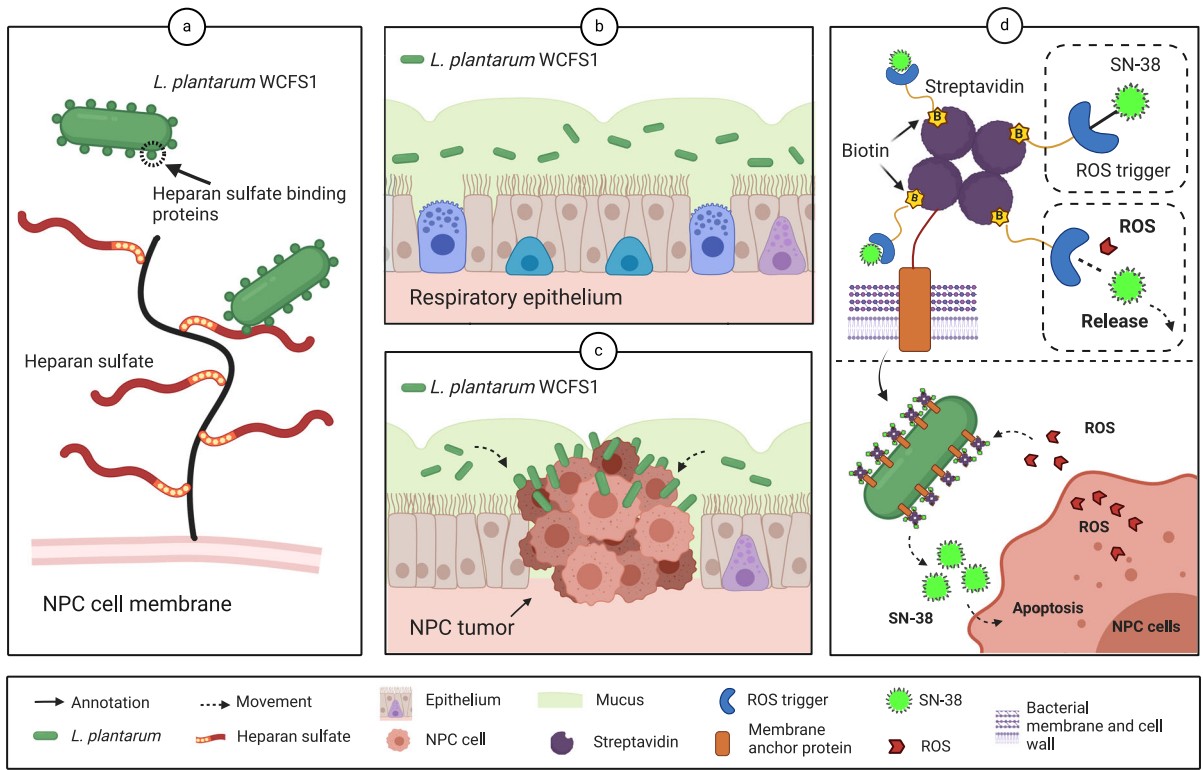

**Fig. 1 | Delivery of SN-38 prodrug by engineered Lp to NPC cells. a** Lp recognizes NPC cells via heparan sulfate binding. **b**, **c** Specific binding of Lp to NPCs. **d** Loading of biotinylated prodrugs on Lp and release of SN-38 by Lp near NPC cells.

manner. To further improve selectivity, some prodrugs are combined with tumor-targeting components, macromolecule carriers such as nanomaterials[11,12], and antibodies[13,14] for site-directed delivery of the payload drugs. This is achieved by the recognition of tumor-specific markers, such as antigens or receptors targeted by antibody-conjugated drugs (ADCs). To date, 14 ADCs have received market approval from the US Food and Drug Administration (FDA) for cancer treatment, marking a significant advancement for targeted cancer therapy[13].

Nevertheless, the current prodrug strategies have limited target specificity. For instance, TME cues[15,16] may not be clearly distinguishable from normal tissue, which would result in significant off-target effects in prodrugs. The conjugation of tumor-targeting carriers could substantially improve the treatment specificity, yet the macromolecular nature of the carriers often complicates the pharmacokinetic profiles of the prodrugs in the circulation, affecting their biodistribution, metabolism and clearance. In the case of ADCs, conjugating the antibody carriers to the native drugs substantially increases the size of the molecule, which impairs the penetration and bioavailability of the drugs in the tumors[17]. The longer half-life of the carrier antibodies also prolongs the clearance of payload drugs in the body, damaging liver and kidney functions[18]. In addition to systemic toxicity caused by chemotherapy, systemic administration of ADCs can introduce toxicity from the antibody components, inducing immune responses and causing severe secondary injuries that lead to nephrotoxicity in patients[19]. Several preclinical studies have also revealed similar complications and side effects for various nanomaterial carriers[20].

To overcome the limitation stemming from confined treatment specificity and associated complications, our aim is to harness the intrinsic interactions between bacteria and cancer cells within the tumor microbiome for the precise administration of prodrugs. Various carcinoma tissues, including nasopharyngeal[21], breast[22], ovary[22], and colorectal carcinomas[22,23] are reported to host commensal microbes that form unique cancer microbiota that could potentially impact the carcinogenesis, progression, and treatment of the host tumor cells[24,25]. These findings led us to formulate the hypothesis that we could pinpoint commensal microbes endowed with cancer-binding capabilities and engineer them to precisely deliver prodrugs to specific cancer sites. The payload prodrugs can be conjugated to the bacterial vector through chemical linkers to ensure the site-specific release of the native drugs. To test our hypothesis, we aimed to identify and characterize commensal microbes with the ability to bind to cancer cells and subsequently engineer these microbes to enable the precise delivery of chemotherapy agents to cancer sites, using nasopharyngeal carcinoma (NPC) as a representative model for cancer. NPC is the most common head and neck cancer and has a high incidence rate in Southern China, Southeast Asia, and North Africa, affecting over 133,000 people and causing over 80,000 deaths in 2020 alone[26]. By 2040, it is projected that the global number of NPC cases and deaths will rise to approximately 179,000 and 114,000, respectively[27]. Chemotherapy in NPC is associated with substantial toxicity and difficulties in administration, making prolonged chemotherapy intolerable for patients[28,29].

In this work, we identify and characterize a human oral isolate—the *Lactobacillus plantarum* WFCS1 (Lp) strain—that demonstrates specific binding to NPCs through interactions mediated by the oligopeptide-binding (OppA) protein and heparan sulfate (Fig. 1a–c). We engineer Lp to present tetramer streptavidin on its surface, enabling the loading of biotinylated prodrugs that can undergo bioconversion into the chemotherapy agent SN-38 in close proximity to NPCs (Fig. 1d). The engineered Lp displays substantial synergy with the loaded prodrug, resulting in a reduction of up to 10-fold in the half maximal inhibitory concentration (IC50) required for SN-38. In an NPC mouse xenograft model, the delivery of prodrugs through engineered Lp leads to a 67% inhibition in tumor growth, significantly enhancing the efficacy of SN-38 by 54%.

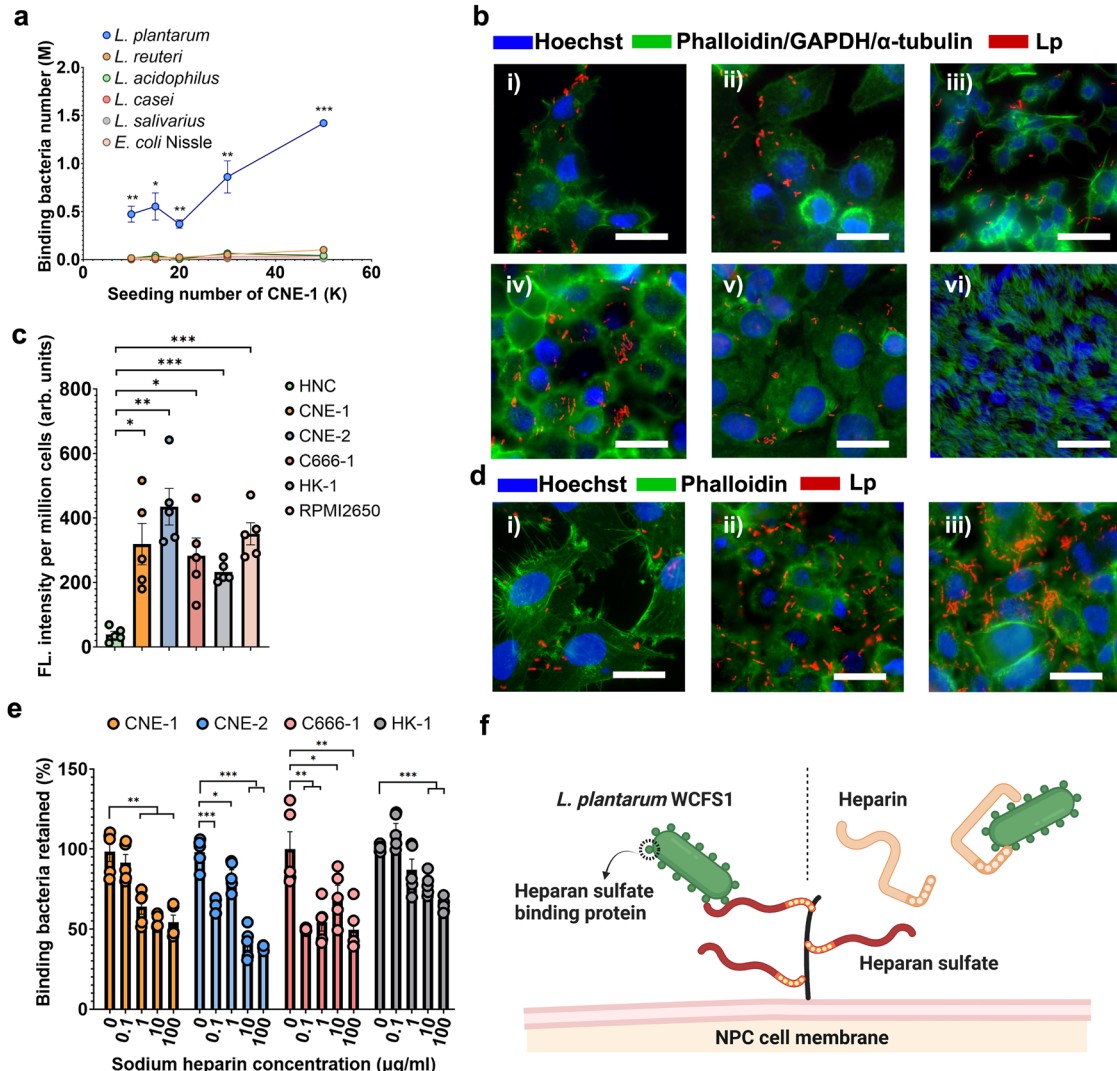

**Fig. 2 | Characterizing the binding affinity of *L. plantarum* WCF1 toward NPC cells. a** Quantification of the cancer binding capacity of commensal strains using CNE-1 as the binding target. n = 3 cell cultures. (*P* values vs *L. reuteri* at 10k seeding number = 0.0053, at 15k seeding number = 0.0193, at 20k seeding number = 0.0013, at 30k seeding number = 0.0085, at 50k seeding number = 3.23 × 10⁻⁶). **b** Lp localized on (i) CNE-1, (ii) CNE-2, (iii) C666-1, (iv) HK-1, and (v) RPMI2650 cell surfaces and absent on (**vi**) HNC cells. Blue - mammalian nucleus stained by Hoechst. Green - cell cytoskeleton stained by phalloidin, cell cytoplasm stained against GAPDH (RPMI 2650 cells), or microcilia stained against α-tubulins (HNC cells). Red - Lp expressing mCherry. Scale bar -- 25 μm. **c** Quantification of Lp-cell binding affinity. The y-axis indicates the fold change in the FITC fluorescence signal, standardized to that of the HNC group. n = 5 experimental replicates. (*P* values HNC vs CNE-1 = 0.0025, vs CNE-2 = 0.0001, C666-1 = 0.0024, vs HK-

1 = 3.25 × 10⁻⁶, vs RPMI2650 = 2.27 × 10⁻⁵). **d** IF staining showing interactions between Lp and (i) T24 bladder, (ii) A549 lung, and (iii) AGS gastric cancer cells. Scale bar -- 25 μm. **e** Inhibition of Lp-NPC binding by heparin salts in the coculture of Lp and four NPC cell lines. n = 5 experimental replicates. (*P* values in CNE-1 group 0.1 ug/ml vs control = 0.4144, 1 μg/ml vs control = 0.0021, 10 μg/ml vs control = 0.0001, 100 μg/ml vs control = 0.0004; in CNE-2 group 7.06 × 10⁻⁵, 0.0289, 1.08 × 10⁻⁵, 4.58 × 10⁻⁷; in C666-1 group 0.0016, 0.0054, 0.0499, 0.0038; in HK-1 group 0.0568, 0.0816, 0.0001, 2.45 × 10⁻⁷). **f** Schematic diagram showing possible competitive inhibition between heparan sulfate and heparin. Error bars indicate the standard error (SEM). All the data are presented as mean values ± SEM. Unpaired two-sided Student's *t* tests were performed to determine the statistical significance. *P < 0.05; **P < 0.01; ***P < 0.001. Source data are provided as a Source Data file.

## Results

### Identification and characterization of the NPC binding strain *L. plantarum* WCFS1

To identify a microbe that possesses cancer-binding capabilities, we investigated six commensal strains, including *Lactobacillus plantarum* WCFS1 and *Lactobacillus casei* DSM 20011, *Lactobacillus acidophilus* DSM 20079, *Lactobacillus reuteri* DSM 20016, *Lactobacillus salivarius* DSM 20555, and an *Escherichia coli* Nissle 1917 strain. The strains were selected based on their prevalence within their higher species abundance in the nasopharynx[30] and NPC[21], as well as their applications in otolaryngology[31–35]. We tested the binding affinity of these strains toward NPC using the human NPC cell line CNE-1 as the target. As shown in Fig. 2a. *L. plantarum* WCFS1 (Lp) exhibited the highest

binding affinity toward CNE-1 cells compared to the other bacteria and showed a positive correlation with the confluence of the CNE-1 cells. When applied to a monolayer of CNE-1 cells, with 50 thousand cells seeded, Lp bound to CNE-1 cells at a ratio of approximately 9 bacteria per CNE-1 cell, which was 14 times higher than that of the second strongest binder, *L. retureri* (Supplementary Fig. 1).

We then tested the binding efficacy of Lp to cancer cells and noncancerous cells of human nasal origin to further evaluate its specificity toward NPC. To visualize the binding, we engineered Lp to express a red fluorescent protein (Supplementary Fig. 2) and cocultured the engineered strain with four NPC cell lines, including CNE-1. CNE-2, C666-1 and HK-1, a nasal squamous cell carcinoma line RPMI 2650, and a human healthy nasal cell line (HNC). Immunofluorescence

staining (IF) of the Lp-NPC coculture showed that Lp strain bound to all cancer cells but not HNC cells (Fig. 2b). Through scanning electron microscopy (SEM), we found that Lp was present on the surface of CNE-1 cells in coculture and was not internalized by the host cells (Supplementary Fig. 3). In addition, we labeled Lp with fluorescein isothiocyanate (FITC) and quantitatively compared the bacterial binding efficacy toward different cells. The binding efficacy of Lp to various cancer lines was 6 to 20 times higher than that to HNCs (Fig. 2c). Lp was also found to interact with bladder, lung, and gastric cancer cells (Fig. 2d).

NPCs are known to overexpress heparan sulfate proteoglycans and have exposed heparan sulfate on the cell surface[36,37], which has been reported as a binding target for *Lactobacillus* adhesins[38]. The respiratory epithelium in the nasal cavity does not present apical heparan sulfate[39], and ciliated HNCs that mimic the nasal epithelium[40] display negligible binding. For these reasons, we hypothesized that heparan sulfate serves as one of the binding targets in the Lp-NPC interaction. To validate this hypothesis, we incubated cancer cells with a heparan sulfate analog, heparin, prior to coculture. The addition of sodium heparin salt (HS) significantly inhibited Lp-NPC binding (Fig. 2e), which may be explained by the competitive inhibition of HS to the binding between Lp and heparan sulfate (Fig. 2f). At the highest concentration, HS reduced the binding of Lp to all four NPC cell lines by 35–60% (Fig. 2e). Taken together, these data suggest that Lp can bind to NPCs specifically with high affinity, and heparan sulfate on NPCs may be the potential binding target for Lp.

## Elucidation of bacterial adhesin mechanisms for NPC binding

To elucidate the molecular mechanisms underlying the binding of Lp to heparan sulfate on cancer cells, we hypothesized that oligopeptide binding proteins (OppAs) within Lp might play a contributory role in facilitating Lp-NPC binding. This hypothesis has been formulated based on the subsequent findings. First, the adherence of *L. salivarius* to HeLa cells is mediated through the interaction between the bacterial OppA protein and glycosaminoglycans on the eukaryotic cell surface[38]. Second, upon conducting a sequence alignment of the OppA protein derived from *L. salivarus*, we identified the presence of seven homologous copies of OppA proteins within the Lp genome (Fig. 3a, b). This count surpasses the number of OppA protein copies reported in other *Lactobacillus* strains employed in this study[41].

To investigate the interaction between OppA and heparan sulfate, we predicted the three-dimensional structure of all OppA proteins through AlphaFold[42] and performed heparin docking on the simulated protein models using Cluspro[43] with tetramer heparin as the ligand. Based on the docked complex analysis, all proteins are predicted to have potential heparin-binding capability and display two common heparin binding sites (Supplementary Fig. 4). Among the proteins, Lp_0783, Lp_0201 and Lp_1261 were predicted to have only one of the heparin binding sites, while Lp_0018, Lp_0092, Lp_0200 and Lp_3686 were shown to have both heparin binding sites (Supplementary Fig. 4, Supplementary Table 2). Among the OppA proteins, Lp_0018 was predicted to have much lower heparin binding energy at both sites, indicating a stronger interaction with the ligand heparin (Fig. 3c, Supplementary Fig. 5, Supplementary Tables 1 and 2). From the superimposed docked complexes and sequence alignment, it was clear that contact map regions were highly conserved among the different OppA proteins (Supplementary Fig. 6). However, Lp_0018, in comparison to other OppA proteins, has many more arginine, lysine and asparagine residues, which are known to have strong interactions with heparin and heparan sulfate[44–47], in both binding sites (Fig. 3c and Supplementary Fig. 5), which accounts for a stronger heparin binding efficacy. In addition, the heparin – Lp_0018 docking complex showed that the positions of heparin ligand clusters are proximal to each other and have different orientations in their binding sites (Supplementary Fig. 7). Since heparin and heparan sulfate are polysaccharides in

nature, it is possible that both binding sites are involved in polysaccharide binding. Taken together, Lp_0018 is predicted to have high heparan sulfate binding affinity.

## Characterization of OppA binding efficacy to cancer cells

To examine the binding efficacy of the OppA proteins to heparan sulfate on cancer cells, we recombinantly expressed the OppA proteins (Supplementary Table 3) with a cMyc tag fused to the C terminus for detection (Supplementary Fig. 8a). The recombinant proteins were purified through a heparin affinity column and displayed different binding affinities toward heparin (Supplementary Fig. 9). Among the proteins, Lp_0200 was expressed in a truncated version for better solubilization and tagged with a polyhistidine tag in the N-terminus for alternative purification in the nickel column. Through flow cytometry analysis, we found that all OppA proteins bound to all five cancer cells, where Lp_0018 exhibited the highest binding intensity and was a universal binder for all cancer cells, and Lp_0092 presented the weakest binding protein (Fig. 3d and Supplementary Fig. 8). To further validate the Lp-cancer binding, we incubated Lp_0018 with both NPC and HNC cells and observed the OppA-NPC binding through IF staining. As shown in Supplementary Figs. 10, 11, all four NPCs express heparan sulfate proteoglycans (HSPG) syndecan-1 and syndecan-2 which are absent in HNC cells. Correspondingly, Lp_0018 was shown to bind to the NPCs, but not the HNC cells, aligning with the distribution of syndecan-1. These results are in accordance with the prediction from OppA-heparin docking simulations.

To further verify the role of Lp_0018 in Lp-NPC binding, we attempted its deletion using a CRISPR-Cas9 system previously described for *Lactobacillus*[48]. However, transformation with the editing plasmid resulted in severe growth inhibition, rendering the deletion of Lp_0018 unsuccessful (Supplementary Fig. 12). As OppA proteins are vital for bacteria to acquire oligopeptides[41], the loss of Lp_0018 is likely detrimental to the Lp strain.

Consequently, we further validated the function of OppA proteins by applying them in the coculture of Lp and NPC cells. We hypothesized that the recombinant OppA proteins would compete for the same binding sites on NPC with Lp and hence reduce Lp-NPC binding. However, when the strongest two binding proteins, Lp_0018 and Lp_0200 (Fig. 3d i), were added to the Lp-CNE-1 coculture, the bacteria bound to CNE-1 cells were increased by 27- and 2-fold, respectively (Supplementary Fig. 13a). We then examined whether such enhancement in binding is universal and could be observed in other cell lines. When added to the coculture, we found that Lp_0018 increased the bound bacteria in all NPC cells, and the increase was positively correlated with the concentration of Lp_0018 (Fig. 3e). At a concentration of 10 μg/ml, Lp_0018 increased the bound bacteria on various NPC cells by 8- to 27-fold (Fig. 3e). This finding was unexpected and contradicts a previous study on OppA proteins in *L. salivarius*[38]. Interestingly, the addition of heparin salt in the coculture diminished the binding enhancement by Lp_0018 (Fig. 3e), which suggested that heparin blocked the binding of Lp_0018 to heparan sulfate on cancer cells.

Notably, Lp_0200 had a very limited effect on enhancing Lp-CNE-1 binding compared to Lp_0018 (Supplementary Fig. 13a), although both proteins themselves exhibited high binding efficacy toward CNE-1 cells (Fig. 3d i). When incubated with Lp alone, both proteins were shown to adhere to the surface of the bacteria, and Lp_0018 exhibited much stronger adherence toward Lp than Lp_0200 (Supplementary Fig. 13b). This prompted us to hypothesize that the OppA proteins enhance Lp-NPC binding by adhering to Lp and NPCs simultaneously, and the weak bacteria binding capacity of Lp_0200, likely due to the loss of the N-terminal sequences, limited its capacity to enhance Lp-CNE-1 binding (Fig. 3f). To verify this, we removed the N-terminal sequences of Lp_0018 and recombinantly expressed its substrate binding domain -- Lp_0018 SBD. The loss of N-terminal sequences in Lp_0018 SBD demolished its binding capability to Lp (Fig. 3g), and incubation of

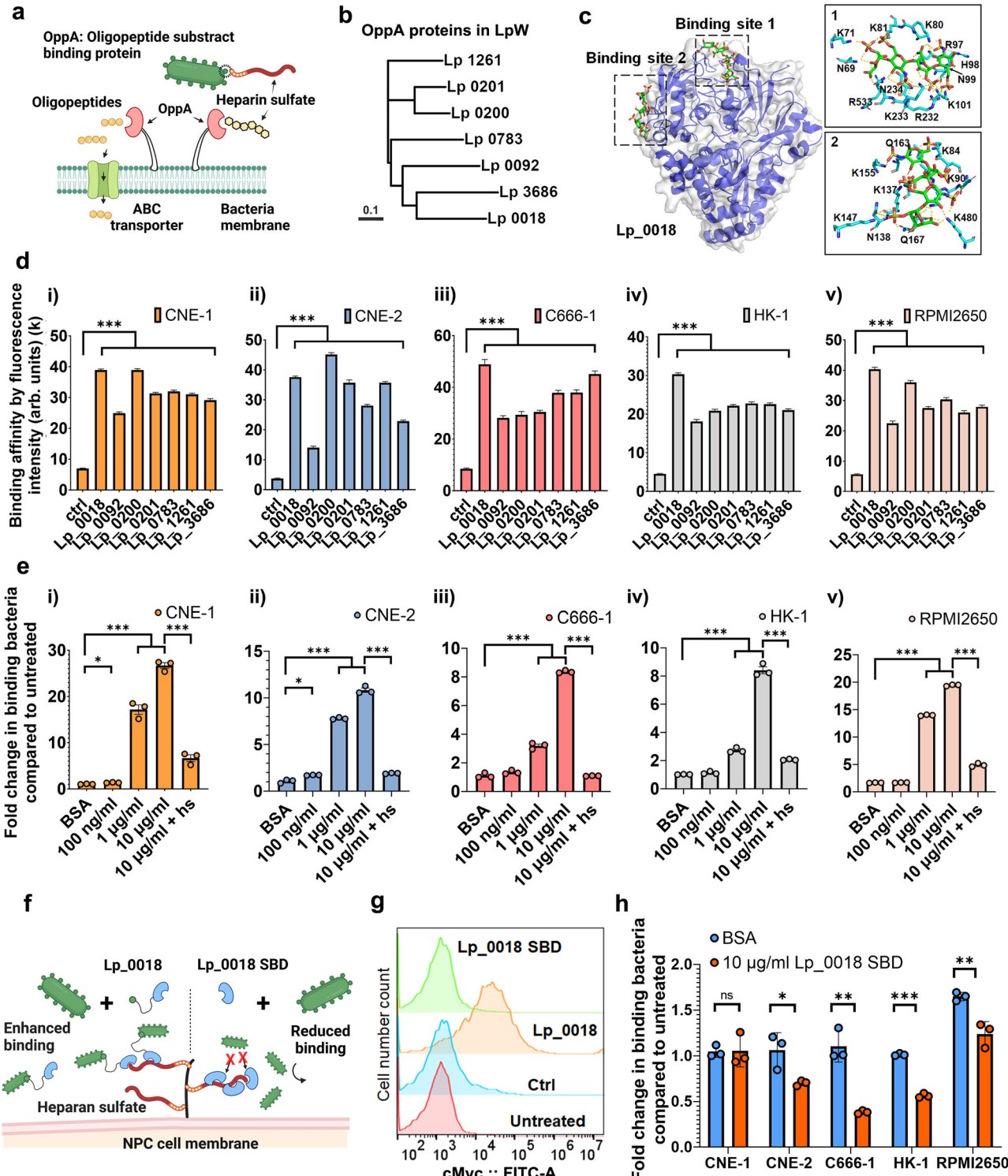

**Fig. 3 | Identification of cancer binding protein from *L. plantarum* cell surface.**
**a** Schematics showing OppA protein binding to heparan sulfate. **b** Polygenetic trees of OppA proteins in Lp. The scale bar indicates 0.1 amino acid replacements per site. **c** Alpha-fold prediction model of Lp_0018 and the simulation of its heparin docking complex. **d** Binding affinity of OppA proteins toward all NPCs by flow cytometry analysis. n = 10,000 events. Data are presented as geometric means ± SEM. (All *p* values < 0.0001. Detailed calculations *p* values are provided in the Source Data file). **e** Increased Lp-NPC binding by the addition of Lp_0018. n = 3 experimental replicates. Data are presented as mean values ± SEM. (*P* values in CNE-1 group 100 ng/ml vs BSA = 0.0047, 1 μg/ml vs control = $7.48 \times 10^{-7}$, 10 μg/ml vs control = $2.44 \times 10^{-6}$, 10 μg/ml vs 10 ug/ml + hs = $3.52 \times 10^{-5}$; in CNE-2 group 0.0043,

0.0110, $2.79 \times 10^{-6}$, $2.72 \times 10^{-6}$; in C666-1 group 0.1307, 0.0002, $3.63 \times 10^{-7}$, $2.01 \times 10^{-8}$; in HK-1 group 0.1161, $6.88 \times 10^{-5}$, $7.08 \times 10^{-6}$, $1.33 \times 10^{-5}$; in RPMI2650 group = 0.4641, $6.25 \times 10^{-9}$, $3.74 \times 10^{-9}$, $2.35 \times 10^{-7}$). **f** Schematics showing Lp_0018 enhancing and Lp_0018 SBD reducing Lp-NPC binding. **g** Binding of Lp_0018 to the surface of Lp. n = 10,000 events. **h** Reduction of Lp-NPC binding by the addition of Lp_0018 SBD. n = 3 experimental replicates. Data are presented as mean values ± SEM. (*P* values in CNE-1 group = 0.9384, in CNE-2 group = 0.0315, in C666-1 = 0.0020, in HK-1 group = $8.82 \times 10^{-6}$, in RPMI2650 group = 0.0078). Unpaired two-sided Student's t tests were performed to determine the statistical significance. *P < 0.05; **P < 0.01; ***P < 0.001. Source data are provided as a Source Data file.

Lp_0018 SBD did not increase Lp-NPC binding in the coculture of any cell lines (Fig. 3h). This proves that recombinantly expressed Lp_0018 binds simultaneously to the host bacteria and NPCs. In addition, in four of cell lines, the addition of Lp_0018 SBD reduced Lp-NPC binding, varying from a 25% reduction in RPMI2650 cells to a 65% reduction in C666-1 cells (Fig. 3h). The absence of reduction in CNE-1 cells suggests additional mechanisms in Lp-CNE-1 binding other than Lp_0018. Accordingly, we conclude that OppA proteins are a major effector in the binding between Lp and NPCs, and heparan sulfate is the binding target on NPCs for OppA proteins.

## Development of a prodrug-carrying Lp strain

Leveraging the cancer-binding characteristic of Lp, we genetically modified Lp to serve as a carrier strain for the delivery of prodrugs to cancer cells. To enable genetic engineering in Lp, we characterized a library of promoters in Lp using plasmid pTRK-892 (Supplementary Fig. 14a) based on previous studies (Supplementary Table 4). However, we found that plasmid pTRK-892 was unstable in *E. coli*, and cloning for secreted proteins often leads to loss-of-function mutations. To ease the difficulty in cloning, two new plasmids, pHSSC256 and pHSBR256, were created based on a narrow-host-range p256 replicon[49] (Supplementary Fig. 14b, c).

We chose streptavidin (Sav) as the carrier protein for prodrugs due to the high binding affinity and specificity between tetramer streptavidin and biotinylated molecules[50]. Three different Sav fusion proteins were cloned into pHSSC256 with a cMyc tag at the C-terminus. The N-terminus was fused to three surface-displaying sequences: Lp_1452, Lp_1568, and Lp_3014 natively from Lp[51] (Supplementary Fig. 14d). Through flow cytometry analysis and IF staining, streptavidin was detected on the surface of Lp after its fusion to Lp_1568 (Lp_1568 – Sav) (Supplementary Fig. 14e, f). We further engineered an Lp strain displaying homotetrameric streptavidin (Lp-Sav) by combining Lp_1568 – Sav with a streptavidin secretion cassette and a monomer streptavidin control strain (Lp-mSav) with an intracellular streptavidin expression cassette (Fig. 4a, b). We detected Lp_1568-Sav in both Lp-mSav and Lp-Sav strains in the insoluble fraction through the detection of the cMyc tag by western blotting (Fig. 4c), flow cytometry analysis, and IF staining targeting (Fig. 4d). On the other hand, excessive Sav proteins were detected only in the culture supernatant of the Lp-Sav strain, which could interact with Lp_1568-Sav and multimerize into the tetramer streptavidin (Fig. 4c). As shown in Fig. 4e, the Lp-Sav strain displayed HA-tagged streptavidin on its surface, possibly in the form of multimer streptavidin (Fig. 4e). We then tested the functionality of Lp-Sav through the loading of a biotinylated fluorescent probe – atto-565 (Fig. 4f). Through direct incubation in PBS, we observed nonspecific attachment of atto-565 to the empty vector (EV) strain, likely due to native biotin-binding proteins present in *Lactobacillus*[52,53]. Conversely, the Lp-Sav strain demonstrated significantly higher loading of atto-565 compared to the EV strain (Fig. 4f). Taken together, we successfully engineered Lp to display an active tetramer streptavidin protein on its bacterial surface and enabled the surface loading of biotinylated molecules in the engineered strain.

## Loading and controlled release of prodrugs

To enable the loading and controlled release of prodrugs in Lp-Sav, we devised a strategy involving the design and synthesis of two biotinylated derivatives of SN-38 (referred to as SN hereafter). These prodrugs were constructed with two established redox stress-responsive linkers: a thioketal linker[54–56] (TL-SN) and a boronic ester linker[57,58] (BL-SN) (Fig. 4g and Supplementary Fig. 15a). When excited at 363 nm, both prodrugs generate distinctive emission peaks at 445 nm, which helps to differentiate the prodrugs from native SN (Fig. 4h and Supplementary Fig. 15b). When incubated with the Lp-Sav strain, TL-SN could be loaded onto the bacteria at the highest concentration of 8.9 μM (Fig. 4i). Similar to the loading of atto-565, nonspecific loading of TL-SN

in the EV strain was also observed, with a saturated loading concentration of 4.6 μM (Fig. 4i). For BL-SN, the loading of BL-SN in both Lp-Sav and EV strains was positively correlated with the incubation concentration and did not reach saturation in the highest concentration incubated (Supplementary Fig. 15c). The nonspecific loading of BL-SN in the EV strain is 10 times greater than that of TL-SN, which might be explained by the strong complexation ability of borate to hydroxy groups on the bacterial surface[7,59]. Moreover, Transmission Electron Microscopy (TEM) analysis revealed that the incorporation of both prodrugs did not induce any morphological changes in Lp (Supplementary Fig. 16).

As illustrated in Fig. 5a, b, TL-SN permits the release of free SN in response to both reactive oxygen species (ROS) and glutathione (GSH). BL-SN, however, is only activated by ROS (Supplementary Fig. 17a, b). We initially tested the stability of TL-SN and BL-SN-loaded Lp-Sav strains (referred to as Lp-Sav-TL-SN and Lp-Sav-BL-SN, respectively) in PBS without any activators. Both Lp-Sav-TL-SN and Lp-Sav-BL-SN demonstrated minimal activation and subsequent release of SN over a 15-h incubation period (Supplementary Fig. 18). We then incubated them with the activators $H_2O_2$ or GSH to test the discharge of SN from Lp-Sav. Interestingly, compared to GSH, TL-SN was more sensitive to $H_2O_2$ and released SN at a higher reaction rate in $H_2O_2$ (Fig. 5c, f). Moreover, the byproducts of TL-SN decomposition generated an emission peak at 420 nm, which contributed to the background fluorescent reading of TL-SN at 445 nm (Fig. 5d, g). Nevertheless, we measured the concentration of TL-SN and SN according to their fluorescent signal and found that more than 80% of TL-SN was converted to SN 5 or 10 h after incubation with $H_2O_2$ or GSH (Fig. 5e, h).

On the other hand, the activation of BL-SN by $H_2O_2$ led to rapid lysis of the bacteria and the quenching of BL-SN fluorescence (Supplementary Fig. 17c, d). Notably, cell lysis is more significant at subphysiological pH values (Supplementary Fig. 17c) closer to the extracellular pH of cancer cells[7]. Following cell lysis and the quenching of BL-SN fluorescence, we detected a gradual release of SN in the supernatant, which peaked at 24 h of incubation (Supplementary Fig. 17e).

We then cocultured the prodrug-loaded Lp-Sav strains, Lp-Sav-TL-SN and Lp-Sav-BL-SN, with C666-1 cells without the addition of activators to further investigate the discharge of SN in vitro. Both Lp-Sav-TL-SN and Lp-Sav-BL-SN were found to release SN, which led to the accumulation of SN in C666-1 cells (Fig. 5i and Supplementary Fig. 17f). Previous reports have indicated that NPC tumors from patients exhibit high levels of oxidative stress and GSH[60,61]. Our findings suggest that prodrug-loaded Lp-Sav strains can be activated by these intrinsic activators in NPC cells, leading to the release of SN for targeting the cancer.

## Evaluation of the treatment efficacy of prodrug-loaded Lp-Sav in NPC cell lines

We further investigated the in vitro anticancer efficacies of Lp-Sav-TL-SN and Lp-Sav-BL-SN. We cocultured four NPC cell lines with Lp-Sav-TL-SN and Lp-Sav-BL-SN and measured their 24-h half-maximal inhibitory concentration ($IC_{50}$). To mitigate nutrient competition from bacteria, which can impede cancer cell growth, we supplemented bacteriostatic antibiotics erythromycin and chloramphenicol in the media to restrain bacterial growth during coculture. We observed that when the Lp-Sav strain was cocultured at an initial density up to $OD_{600}$ ~ 0.4, the 24-h coculture period had no discernible impact on the viability of the NPC cells (Fig. 5j). Subsequently, we determined the 24-h half-maximal inhibitory concentration ($IC_{50}$) of prodrug-loaded Lp-Sav strains, utilizing the highest bacterial density at OD600 ~ 0.4.

The treatment with Lp-Sav-TL-SN substantially enhanced the potency of SN, resulting in a 5- to 10-fold reduction in the $IC_{50}$ compared to SN alone in CNE-1, HK-1, and C666-1 cells (Fig. 5k, l). The limited reduction observed in CNE-2 may be attributed to the higher

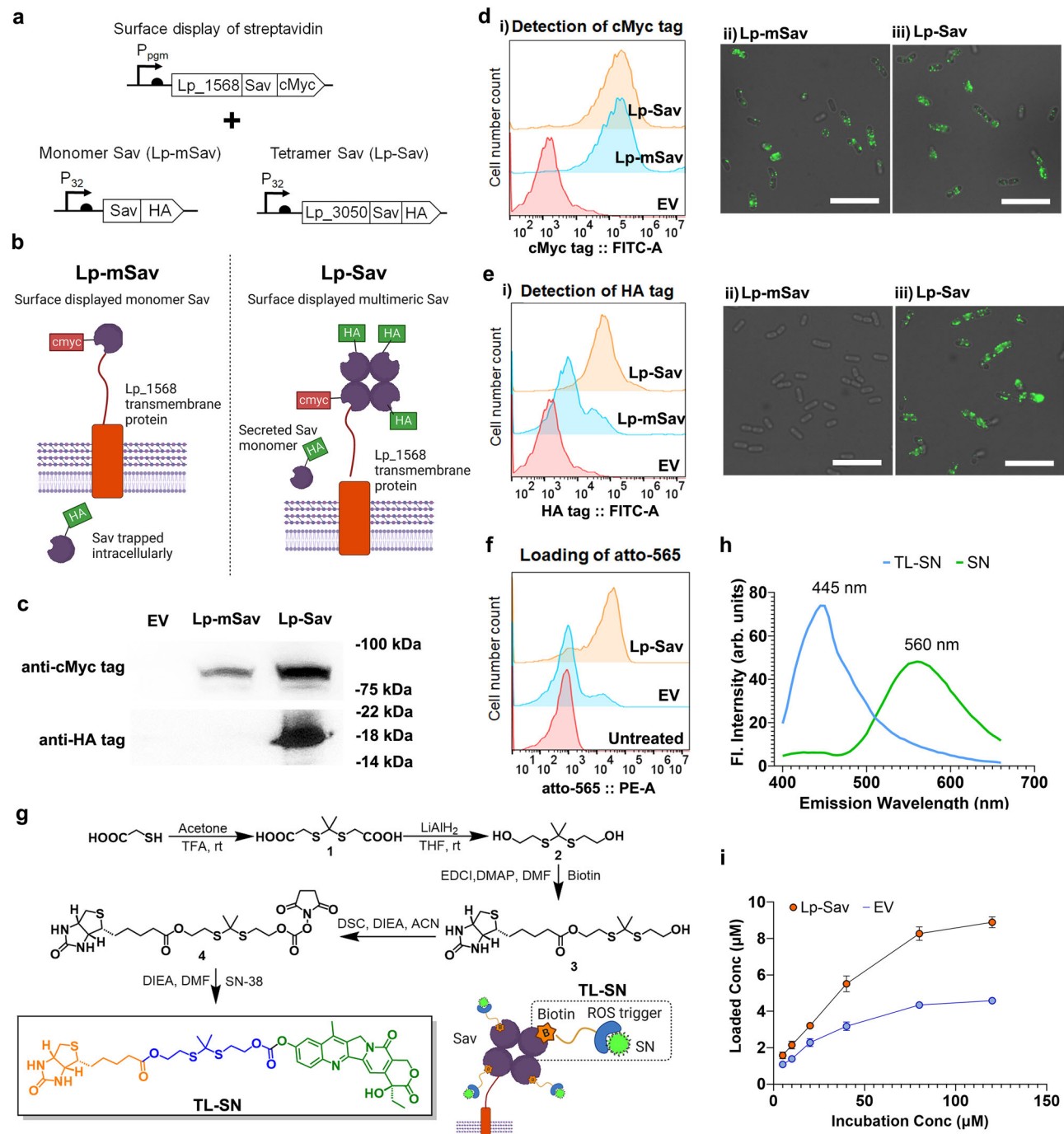

**Fig. 4 | Surface display of streptavidin and loading of the prodrug on *L. plantarum* WCFS1. a** Expression cassette for Sav surface display in Lp. **b** Schematics of the tetramer Sav surface display. **c** Western blot images showing the surface display of the Lp_1568-Sav fusion protein and secreted Sav protein. **d** Detection of Lp_1568-Sav fusion protein on the Lp cell surface. (i) Flow cytometry analysis and (ii) IF. Scale bars, 25 μm. **e** Detection of multimeric Sav on the Lp cell surface via (i) flow cytometry analysis and (ii) IF. Scale bars, 25 μm. **f** Loading of a biotinylated fluorescent probe on the Lp cell surface. **g** Schematics showing the synthesis route of TL-SN and the loading of TL-SN. **h** Fluorescence spectrum scanning showing the optimal emission wavelengths of TL-SN and SN. **i** Loading of TL-SN on the Lp-Sav and EV strains. n = 3 bacterial cultures. Data are presented as mean values ± SEM. Source data are provided as a Source Data file.

resistance of CNE-2 cells to SN. TL-SN exhibited IC50 values similar to SN in the treatment of most NPC cells and nearly 2 times the IC50 of SN in treating CNE-2 cells but the introduction of Lp-Sav improved treatment efficiency of TL-SN in all NPC cells, indicating a synergistic effect between Lp-Sav and TL-SN (Fig. 5k, l). The treatment with Lp-Sav-BL-SN, in contrast, only led to a limited reduction of IC$_{50}$ in CNE-1 and CNE-2 cells (Supplementary Fig. 17g, h). It is conceivable that the bacterial lysis induced by BL-SN activation might have hindered the potential

synergy between Lp-Sav and BL-SN (Supplementary Fig. 17i). These results indicate that Lp-Sav-TL-SN stands as a more potent therapeutic option in comparison to the unmodified drug SN and warrant further evaluation in in vivo studies.

### Biodistribution of Lp in a xenograft NPC mouse model
To evaluate the in vivo anticancer efficacies of Lp-Sav-TL-SN, we first determined whether and the extent to which Lp could bind to cancer

cells in vivo. We established a xenograft NPC mouse model in BALB/c nude mice through subcutaneous injection of C666-1 cells. Subsequently, we administered Lp cells through intravenous injection, and

evaluated the biodistribution of the Lp strain in tumor-bearing mice (Fig. 6a). Specifically, Lp was engineered to express the bioluminescence protein CBluc[62] (referred to as Lp-CB) and was injected through

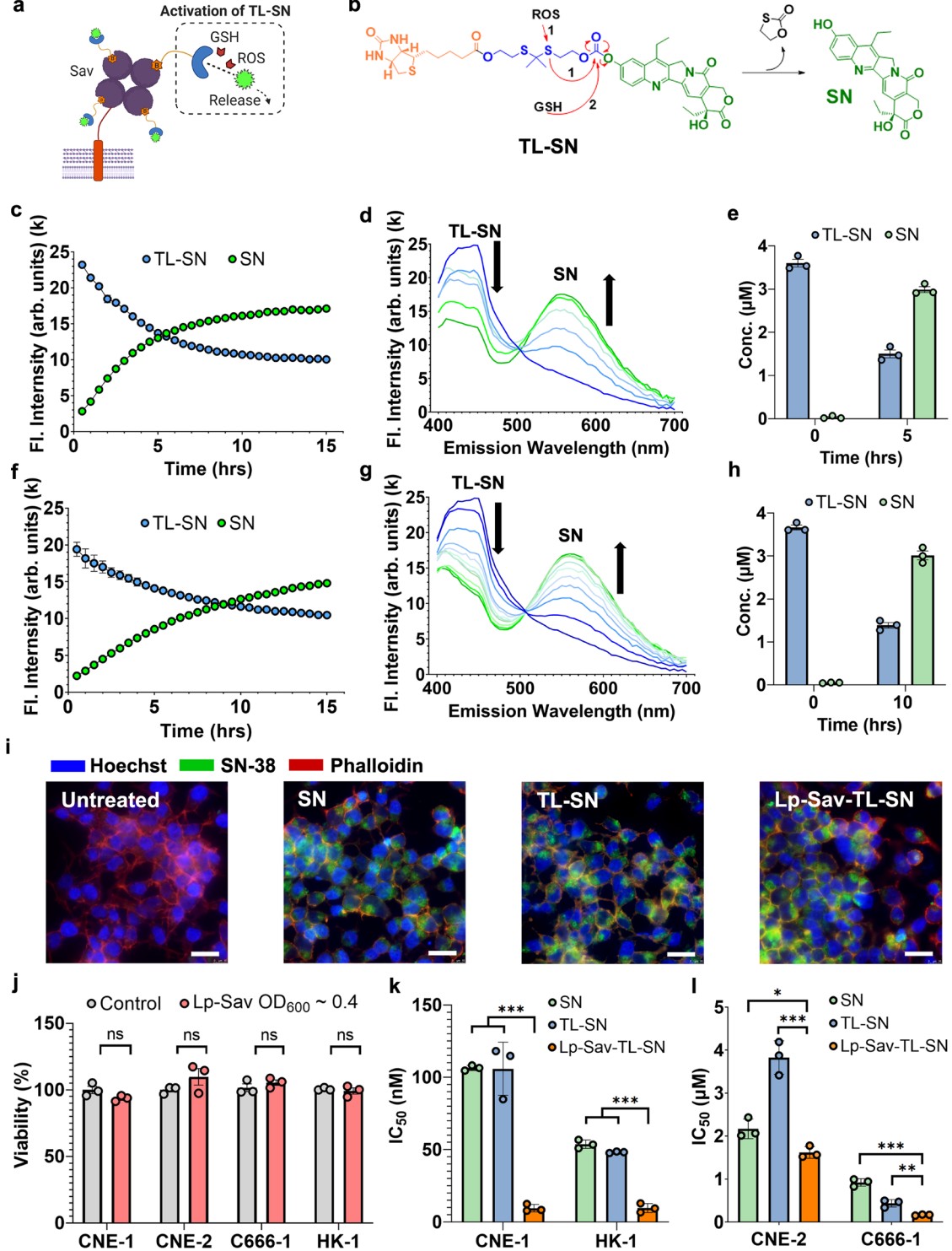

**Fig. 5 | Release of prodrug 1 from Lp-Sav and in vitro characterization of prodrug-loaded Lp-Sav in NPC cells. a**, **b** Schematics showing the mechanism of TL-SN activation and SN release. **c**, **f** Activation of TL-SN and release of SN in TL-SN-loaded Lp-Sav over time. **d**, **g** Dynamics of TL-SN activation by spectrum scanning over time. **e**, **h** Concentration of TL-SN in the cell pellet and SN in the supernatant pre- and postactivation. **c–e** The activation of TL-SN via $H_2O_2$; (**f**), (**g**), (**h**) - the activation of TL-SN via GSH. (**i**) Accumulation of SN in C666-1 cells after TL-SN activation: 24 h incubation time. Red – β-actin stained by phalloidin. Blue – nucleus

stained by Hoechst. Green – SN. Scale bars, 25 μm. **j** Viability of NPC cells after 24 h of coculture with unloaded Lp-Sav at $OD_{600}$ ~ 0.4. **k**, **l** $IC_{50}$ of various treatments in NPC cells. The assays were performed with three biological repeats. All data are presented as mean values ± SEM. (*P* values in CNE-1 group SN vs Lp-Sav-TL-SN = $7.19 \times 10^{-7}$, TL-SN vs Lp-Sav-TL-SN = 0.0008; in HK-1 group $5.66 \times 10^{-5}$, $2.91 \times 10^{-5}$; in CNE-2 group 0.0233, 0.0008; in C666-1 group 0.0001, 0.0058). Unpaired two-sided Student's t tests were performed to determine the statistical significance. *$P < 0.05$; **$P < 0.01$; ***$P < 0.001$. Source data are provided as a Source Data file.

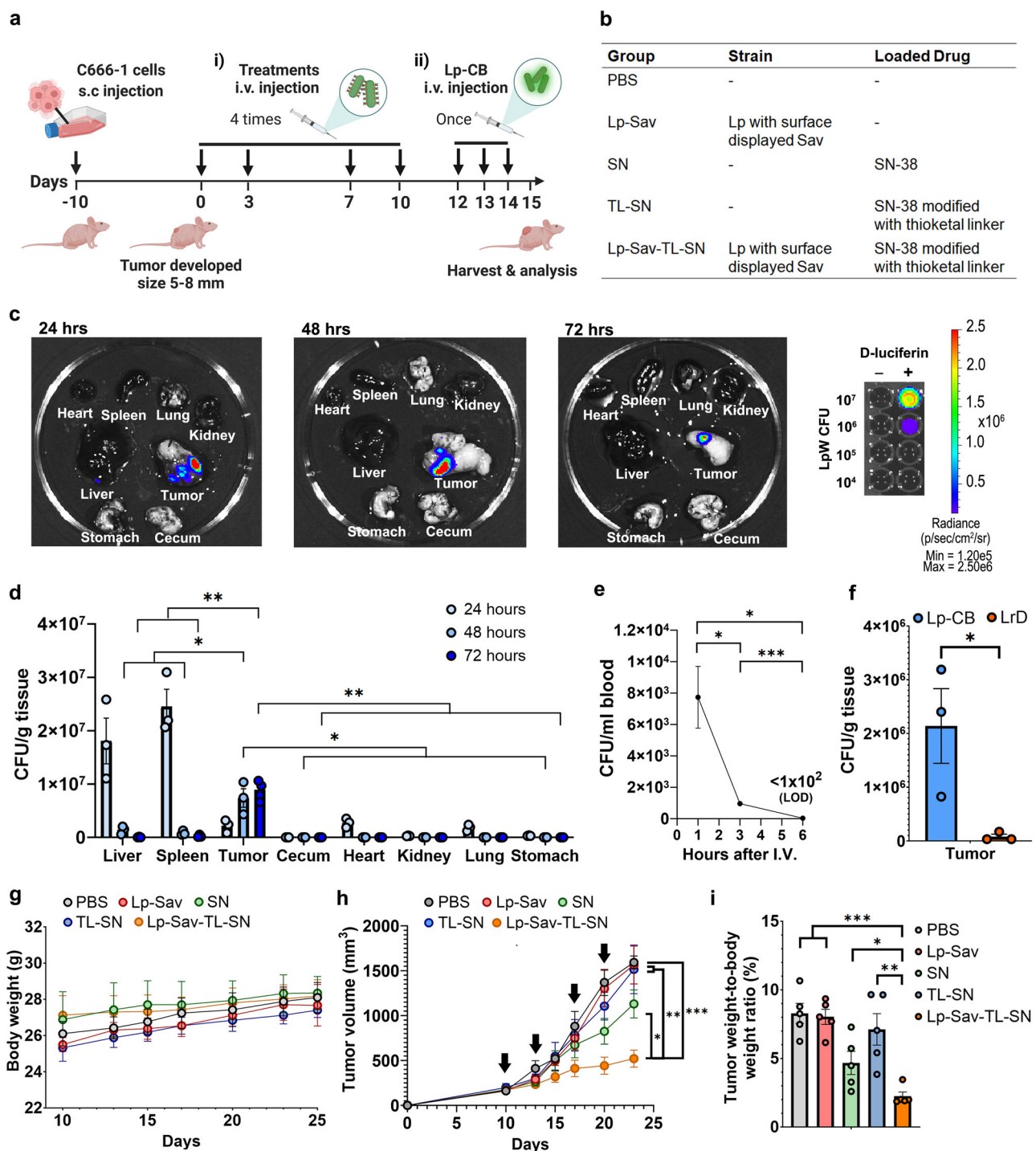

**Fig. 6 | Treatment of the xenograft mouse NPC model. a** Schematics – NPC xenograft mouse model for (i) the evaluation of treatment effectiveness and (ii) the study of Lp biodistribution. **b** Table of the treatment groups used in this study. **c** Visualization of bioluminescent Lp in mouse organs at 24, 48, and 72 h post i.v. injection. **d** Bacteria density of Lp-CB in various mouse organs, including liver, spleen, tumor, cecum, heart, kidney, lung and stomach, 24, 48, and 72 h after injection. n = 3 mice. (P values 48 h post injection tumor vs liver = 0.0290, tumor vs spleen = 0.0205, tumor vs the rest organs = 0.0135; 72 h post injection tumor vs liver = 0.0019, tumor vs spleen = 0.0021, tumor vs the rest organs = 0.0018). **e** Bacteria density of Lp-CB in blood 1, 3 and 6 h after injection. n = 3 mice. (*P* value

1 h vs 3 h = 0.0261, 1 h vs 6 h = 0.0172, 3 h vs 6 h =0.00029). **f** Bacteria density of Lp-CB and LrD in tumor 24 h postinjection. n = 3 mice. (*P* value = 0.0417). **g** Body weight change of the mice in different groups during treatment. n = 3 mice. **h** Tumor progression of mice in various treatment groups. The black arrow indicates injection. n = 5 mice. (*P* value vs PBS = 5.44 × 10⁻⁵, Lp-Sav = 0.0019, SN = 0.0104, TL-SN = 0.0084). **i** Endpoint tumor burden of mice in various treatment groups. n = 5 mice. (*P* value vs PBS = 6.13 × 10⁻⁵, Lp-Sav = 1.62 × 10⁻⁵, SN = 0.0270, TL-SN = 0.0034). All data are presented as mean values ± SEM. Unpaired two-sided Student's t tests were performed to determine the statistical significance. *P < 0.05; **P < 0.01; ***P < 0.001. Source data are provided as a Source Data file.

the tail vein into mice bearing notably large tumors. We then euthanized the mice 24, 48, and 72 h post-injection, followed by organ harvesting for analysis. This was necessary due to the inability of bioluminescence signals generated by Lp-CB to penetrate the mouse skin. Through IVIS analysis, tumors from these mice generated the highest bioluminescence signal among all organs harvested at the same time, which indicated a high amount of Lp-CB localized in tumors following intravenous injection (Fig. 6c and Supplementary Fig. 19). Weak bioluminescence signals were also detected in spleens and livers 24 h postinjection, indicating the presence of Lp-CB in these two organs (Fig. 6c and Supplementary Fig. 19). However, 48 h after injection, no bioluminescence was observed in tissues other than tumors (Fig. 6c, Supplementary Fig. 19).

To quantify the amount of Lp-CB in various organs, we homogenized them and measured the bacterial colony forming unit (CFU) of Lp-CB in them (Fig. 6d). In accordance with the results from IVIS analysis, following intravenous administration, the liver, spleen, and tumor exhibited the highest accumulation of Lp-CB (Fig. 6d). Within 24 h post-injection, both liver and spleen harbored a greater Lp-CB concentration compared to tumors (Fig. 6d). However, the number of Lp-CB colonizing tumors consistently increased over the course of 72 h, while Lp-CB levels in the liver and spleen experienced predominant clearance within 48 h (Fig. 6d). Within the 72-h timeframe, the tumor-to-liver and tumor-to-spleen ratios for Lp-CB escalated, transitioning from being less than 1:6 and 1:11 to surpassing 200:1 and 35:1, respectively (Fig. 6d). Furthermore, Lp-CB strains were also detected in the heart with a density comparable to that in the tumor and significantly lower levels in other organs, 24 h postinjection (Fig. 6d). However, bacteria in these organs were no longer detectable 48 h post-injection (Fig. 6d). The transient presence of Lp-CB in the organs implies circulation of the bacteria in the mouse bloodstream. To assess the risk of sepsis, we determined the duration of Lp-CB presence in the blood by monitoring the colony-forming units (CFU) of Lp-CB at 1, 3, and 6 h following intravenous injection (Fig. 6e). Notably, the level of Lp-CB rapidly declined in the mouse blood and became undetectable six hours post-injection (Fig. 6e).

To rule out the nonspecific localization of bacteria in the tumor, we also evaluated the biodistribution of a control *Lactobacillus* strain, *L. reuteri* DSM 20016 (referred to as LrD), in tumor-bearing mice. Following intravenous injection, LrD primarily accumulated in the liver, with comparatively lower levels in all other tissues, and the lowest concentration was observed within the tumors (Supplementary Fig. 20a). In contrast, 24 h postinjection, the density of Lp-CB in tumor (Fig. 6f) and tumor-to-liver ratio of Lp-CB (Supplementary Fig. 20b) was over 35 times and 39 times higher than that of LrD, respectively, indicating higher specificity and efficiency in colonizing NPC tumors.

### Evaluating the treatment effectiveness of Lp-Sav-TL-SN in the mouse model

Subsequently, we further evaluated the efficacy of Lp-Sav-TL-SN in treating NPCs in a xenograft mouse model. We hypothesized that with our strategy, a lower dosage of the prodrug SN would be required to achieve significant inhibition of tumor growth. Therefore, a deescalating treatment regimen was designed for the animal experiment: the treatments were administered at low doses at 50 μg/kg SN, given twice per week for two weeks, totaling 200 μg/kg SN (Fig. 6a). In comparison, previous studies reported the use of Irinotecan, an FDA approved SN prodrug, at higher total doses ranging from 150 mg/kg to 400 mg/kg in less than four weeks[63–66].

To evaluate the efficacy of Lp-Sav-TL-SN, following the initial development of subcutaneous C666-1 tumors (5–8 mm in size), we administered one treatment group (Lp-Sav-TL-SN) along with four control groups (PBS, Lp-Sav, SN, and TL-SN) to the mice bearing the tumors via intravenous injection, respectively (Fig. 6a, b). For each group, a total of four doses were administered to the mice over a span

of two weeks, and no significant weight reduction or other side effects were observed in the mice (Fig. 6g). Based on tumor volume and the tumor-to-body weight ratio, tumors in mice from the Lp-Sav-TL-SN group were 54% smaller compared to the SN group, 66% smaller than the TL-SN group, and 67% smaller than the PBS control group (Fig. 6h, i, and Supplementary Fig. 21).

In summary, our Lp-Sav-TL-SN strain significantly inhibited the progression of NPC tumors. The NPC recognition of native OppA proteins on the bacterial surface enabled bacteria to bind to the NPC surface, and the loaded TL-SN could release SN near the NPC cells to inhibit cancer growth. In vitro experiments showed that the prodrug-loaded microbes had significantly increased potency of SN-38 against NPC cell lines, up to 10-fold higher. In an NPC mouse xenograft model, the delivery of prodrugs through engineered Lp led to a 67% inhibition in tumor growth, significantly augmenting the efficacy of SN-38 by 54%.

## Discussion

Herein, we present a chemotherapy strategy that capitalizes on the inherent interactions between microbes and cancer cells within the tumor microbiome, thereby enabling the precise administration of targeted prodrugs. Our strategy entails the identification and characterization of a commensal microbe possessing intrinsic cancer-binding capabilities. Subsequently, we engineered this commensal to deliver prodrugs to specific cancer sites, using NPC as our model for cancer. We formulated this strategy based on previous research findings. Firstly, diverse carcinoma tissues, including nasopharyngeal carcinoma, harbor distinct cancer microbiota, highlighting the potential role of commensals in the development and progression of cancer[21–23]. Secondly, bacterial-mediated cancer therapy (BMCT) has been investigated primarily using obligate anaerobes like *Salmonella* and *Listeria* to target hypoxic regions within tumors[65]. Nevertheless, these opportunistic pathogens[67] are non-native to the human microbiome and the cancer microbiome, and they may not necessarily possess cancer-specific binding properties. Thirdly, commensals have been genetically modified to produce and localize enzymes for diet-mediated colorectal cancer chemoprevention[68]. However, enzymatic reactions are applicable only to a limited range of substrates, and the absence of these substrates in other areas constrains the utility of this approach.

Our study elucidates a crucial microbial-host interaction within the tumor microbiome, enabling the precise administration of targeted prodrugs. We identified an *L. plantarum* strain that selectively binds to NPCs through the recognition of heparan sulfate, primarily via its OppA proteins. In tumor-bearing mice, intravenous administration of Lp resulted in the accumulation of bacteria specifically within NPC tumors, with Lp being cleared from non-tumoral tissues 48 h post-injection. This clearance of Lp in non-tumoral organs occurs at a rate much faster than that of other tumor-seeking bacteria used in BMCT, including *Salmonella typhimurium* strains[69], *Listeria monocytogenes*[70], *Pseudomonas aeruginosa*[71] and *E. coli*[72,73]. Diverse carcinoma tissues, including NPC, harbor distinct cancer microbiota, with *Lactobacillus* species constituting a significant component within the microbiome of both carcinoma tissues and the corresponding healthy mucosa[21–25]. Therefore, in comparison to many opportunistic pathogens that are non-native to the human microbiome, *Lactobacillus* strains like Lp are readily available in the tumor microenvironment and serves as a safer choice for BMCT applications. Furthermore, heparan sulfate, the binding target of Lp, is a common marker on the surface of many cancer cells[74–76]. Beyond NPCs, we have also demonstrated Lp's affinity for various cancer cell types, including those present in bladder, lung, and gastric tissues, underscoring its potential role in tumor development and its significant potential for broad applications in cancer therapy.

Of particular significance is that our identified strain, Lp, exhibited a synergistic effect with the prodrug, improving the IC50 of SN by up

to 10-fold in the treatment of NPC cell lines. In a xenograft NPC mouse model, the administration of Lp-Sav-TL-SN resulted in a 67% inhibition of tumor growth, significantly enhancing the efficacy of SN by 50%. In clinical settings, prodrugs have been employed to reduce the side effects and enhance the bioavailability of chemotherapy. However, modifications to chemotherapy reagents can reduce chemotherapy toxicity. Prodrugs, on the other hand, remain inactive against cancers until they are activated, and this delayed release mechanism limits their treatment potency. For example, SN has been utilized as a chemotherapy reagent in the form of a prodrug called irinotecan for the first-line treatment of metastatic colon cancer. Irinotecan exhibits cytotoxicity 100 to 1,000 times lower than the native drug SN[65]. In our study, a reduction in potency was observed in unloaded prodrugs TL-SN and BL-SN, both of which exhibited higher IC50 values against NPC cells in vitro. Treatment with SN-TL in tumor-bearing mice did not result in a significant reduction in tumor size. In contrast, the administration of Lp-Sav-TL-SN led to a drastic decrease in the IC50 and a significant reduction in tumor size compared to TL-SN and SN alone, marking an immediate improvement in treatment potency. Additionally, the amount of SN needed in Lp-Sav-TL-SN to achieve significant tumor inhibition is 750 to 2,000 times lower than that in previous studies[63–66]. These findings suggest that this strategy could be an effective approach for treating cancer with reduced chemotherapy requirements and fewer associated side effects, making it a promising approach to de-escalate chemotherapy regimens.

To broaden the potential applications of our approach, we have reprogrammed Lp to display tetramer streptavidin proteins on the bacterial surface, resulting in the engineered strain known as Lp-Sav. This modification allows for the surface loading of biotinylated prodrugs, offering a versatile platform for designing prodrugs with flexibility in their composition. This adaptability enables the selection of active drugs and release mechanisms tailored to the unique characteristics and sensitivities of individual cancers. Firstly, the surface loading pattern serves to protect the prodrugs from being metabolized by the intricate intracellular biochemistry within the bacteria. This protection expands the scope of potential chemotherapy drugs that can be employed. Secondly, the significant differences between bacterial and cancer metabolites make it possible to design linkers that respond to various cues within the tumor microenvironment, including hypoxia[4,5], acidosis[6,7], and high oxidative stress[8,9]. In our system, the combination of Lp-Sav-TL-SN constitutes a comprehensive therapy that achieves both site-targeted delivery through Lp-Sav and site-specific conversion of TL-SN through the embedded thioketal linker[54]. This integrated system simplifies the design of prodrugs and broadens the range of candidate active drugs, making it a promising and versatile approach in cancer therapy.

Due to the challenges associated with developing an orthotopic NPC model, we conducted tests of our strategy using a xenograft mouse model. Our Lp strain has demonstrated the remarkable ability to selectively bind to NPC cells by recognizing heparan sulfate while showing minimal background binding to healthy nasal ciliated cells. This observation closely mirrors the situation in the nasal cavity, where NPC cell surfaces exhibit elevated levels of heparan sulfate due to the overexpression of HSPG, in contrast to the respiratory mucosa within the nasal cavity, where such overexpression is absent[36,37,39]. Despite this targeted delivery, bacterial-mediated cancer therapy presents several limitations. These include a limited payload loading capacity, interference from native bacteria metabolites, and the potential for infection when administered intravenously. These factors restrict the quantity of bacterial vectors permitted in a single treatment, thus complicating the evaluation of treatment plans in the experimental stage and reducing the likelihood of clinical application. As of now, Bacillus Calmette-Guerin (BCG) remains the sole FDA-approved bacterial therapy for cancer. BCG is typically administered locally via intravesical instillation for the treatment of non-muscle-invasive

bladder cancer[77]. We anticipate that our designer therapeutics can be administered in a manner similar to BCG through the mucosal layer to target NPCs, thereby avoiding systemic exposure to both the carrier bacteria and the chemotherapy agents. This approach holds the potential to mitigate the loss of chemotherapy agents due to first-pass metabolism in the liver and reduce systemic chemotherapy toxicity, observed frequently in the case of ADC, offering a promising avenue for further development.

In summary, our study has identified Lp, a commensal strain capable of selectively binding to NPC cancers through the recognition of heparan sulfate. When administered in a xenograft NPC model, Lp exhibited specific colonization within the NPC tumor, with rapid clearance from non-tumoral tissues. Furthermore, we engineered Lp to serve as a versatile drug delivery carrier by displaying streptavidin proteins on its surface. The resulting Lp-Sav strain demonstrated a synergistic effect when loaded with the prodrug TL-SN, significantly enhancing the potency of SN in treating both NPC cell lines and an NPC mouse model. This strategy holds promise as a potential alternative for reducing the intensity of chemotherapy regimens in the treatment of a wide range of cancers.

## Methods

### Ethical regulations
All animal procedures were conducted under Institutional Animal Care and Use Committee (IACUC) guidelines and in conformity with protocols approved by the NUS IACUC (R21-1009) and all relevant ethical regulations.

### Statistics and reproducibility
All in vitro experiments were repeated at least three times independently. Numbers (n) of samples or replicates are indicated in figure legends. Sample sizes for the animal study were selected based on previous studies[55,56]. Mice were randomly grouped following the initial development of tumors and no data were excluded from the analyses. The data are presented as mean values or geometric mean values ± standard error (SEM). Data were statistically analyzed using unpaired two-tailed Student's t-test. A difference was considered statistically significant when $*P < 0.05$, $**P < 0.01$ or $***P < 0.001$. Bioteck Gen5, IVIS spectrum (PerkinElmer), Leica Las X, CytExpert 2.4 (Beckman), Cluspro 2.0 are used for data collection. Microsoft Excel (Office 365, Version 2402), Graphpad Prism 10, Flowjo V10 and PyMOL (2.5.4) were used for data analysis.

### Cloning, protein expression, and purification
The cloning chassis strain *Escherichia coli* DH5α was cultured in lysogeny broth (LB) or brain heart infusion (BHI) broth at 37 °C and 230 r.p.m. *Lactobacillus plantarum* WCFS1 was grown in De Man, Rogosa, and Sharpe (MRS) at 37 °C in static cultures.

OppA proteins were cloned into pET30b expression vectors and expressed under various conditions (Supplementary Table 3). The bacterial strains were harvested, resuspended in lysis buffer (50 mM Tris-Cl with 0.05% Triton X-100; pH 8.0), and lysed using an EmulsiFlex-C3 homogenizer (Avestin). Expressed OppA proteins were purified by fast protein liquid chromatography (ÄKTA pure™) through a heparin column (HiTrap, GE Healthcare). The purified protein was buffer exchanged in PBS, concentrated, and stored at −80 °C. Pilot-scale protein purification and enzymatic assays were conducted using the SynCTI Foundry.

### Cell culture conditions
RPMI 2650 cells were purchased from CLS (Cell Line Services) Germany. A549, AGS, and T24 cells were purchased from ATCC (American Type Culture Collection), America. HK-1 cells were a kind gift from A/Prof Zhong Yong Liang from the Department of Microbiology & Immunology, NUS. CNE-1 and CNE-2 cells were kindly provided by A/Prof Shen

Han Ming from the Department of Physiology, NUS. C666-1 cells were kindly provided by Dr. Joshua Tay from the Department of Otolaryngology, NUS. Human healthy nasal cells (HNCs) were a kind gift from Prof. Wang De Yun from the Department of Otolaryngology, NUS.

Among the cell lines, CNE-1 and CNE-2 were identified as misidentified cell lines maintained by the International Cell Line Authentication Committee. Previous reports provided detailed authentication of both cell lines, which identified distinctive NPC cell genomes alongside genomic contamination from HeLa cells through short tandem repeat profiling[78]. In this study, we identified CNE-1 and CNE-2 cells expressing HSPG syndecan-1 and syndecan-2, rendering them suitable targets for studying the binding between Lp and NPC cells.

RPMI 2650 cells were cultured and maintained in Eagle's Minimum Essential Medium (EMEM) (Lonza). A549, AGS cells, and T24 cells were cultured and maintained in Dulbecco's Modified Eagle Medium (DMEM) (Lonza). NPC cell lines, including HK-1, CNE-1, CNE-2, and C666-1 cells, were cultured and maintained in Roswell Park Memorial Institute (RPMI) 1640 medium (Gibco). All culture media were supplemented with 10% v/v fetal bovine serum (FBS) (Biowest), 100 U/mL penicillin, and 100 μg/mL streptomycin (Gibco). All cells were maintained at 37 °C in a humidified atmosphere with 5% CO2. The generation and cultivation conditions of HNC cells have been previously described[26]. Mycoplasma testing was performed on cell lines using MycoAlert® Mycoplasma Detection Kit (Lonza) according to the manufacturer's instructions. All cells were tested negative for mycoplasma.

### Immunofluorescence staining

Round cover slides (Paul Marienfeld EN) were planted into a 24-well plate and seeded with NPC cell lines. An overnight culture of Lp expressing mCherry was washed 3 times with PBS, diluted in RPMI 1640 media to OD600 ~ 0.5, and cocultured with NPC cells for one hour. After coculture, cover slides with attached cells were washed three times with PBS to wash away excessive bacteria and then fixed for 10 min at room temperature or overnight at 4 °C with 4% paraformaldehyde (PFA, Sigma) in PBS. NPC cells were stained with Alexa Fluor 488 phalloidin (Abcam, ab176753) at a 1:1000 dilution in 1% BSA to visualize the cell cytoskeleton. For RPMI 2650 cells, the cytoplasm was highlighted through immunofluorescence (IF) staining of GAPDH using a primary anti-GAPDH mouse monoclonal antibody (Cell Signaling, CST #97166) at a 1:500 dilution in 1% BSA. For HNC cells, microcilia were stained through IF staining against acetylated α-tubulin using a primary acetylated α-tubulin rabbit monoclonal antibody (Abcam, ab209348). Cell-bound Lp_0018 was detected using a primary mouse monoclonal anti-cMyc antibody (Thermo Fisher MA1-980) at a 1:500 dilution in 1% BSA. Syndecan-1 and syndecan-2 were identified with a primary rabbit monoclonal anti-syndecan-1 antibody (Abcam ab128936) at a 1:200 dilution in 1% BSA and a primary rabbit polyclonal anti-syndecan-2 antibody (Abcam ab205884) at a 1:200 dilution in 1% BSA, respectively. Appropriate secondary antibodies were used to visualize these structures (CST 8889 S, 4412 S, 4408 S, 8890 S) at a 1:500 dilution in 1% BSA. All cells were stained with Hoechst (Sigma) and analyzed by a Leica DM4000 B fluorescence microscope.

### FITC labeling of bacteria and quantification of bacterial binding

Various bacterial strains were incubated with 0.1 mg/ml fluorescein isothiocyanate (FITC) (Sigma–Aldrich) in PBS. The cells were washed four times with PBS to remove excessive FITC and incubated with NPC cells at an OD600 ~ 0.5 in RPMI 1640 media. For the functional analysis of OppA proteins, Lp_0018, Lp_0018 SBD, and Lp_0200 were added to the Lp-NPC co-culture at a final concentration of 10 μg/ml. After incubation for one hour, the cells were washed three times with PBS to remove nonbinding bacteria and detached from plates by trypsin. Cells were then pelleted by centrifuging at 800 × g for 3 min and resuspended in 200 μl of PBS. The total number of epithelial cells was

counted by a cell counter and calculated. One hundred microliters of cell suspension were transferred into 96-well plates, and the attached *L. plantarum* was quantified in a microplate reader (Synergy H1 from Biotek) recording FITC fluorescence intensity (488 nm excitation and 560 emissions).

### Flow cytometry analysis

NPC cells were scraped using cell scrapers and fixed with 4% PFA. The fixed cells were incubated with the 5 μM OppA proteins overnight at 4 °C and then incubated with a c-Myc monoclonal antibody (Thermo Fisher MA1-980) (1:500 dilution in 1% BSA) overnight at 4 °C. The cells were then incubated with a secondary anti-mouse IgG (Alexa Fluor 488, Cell Signaling 4408) (1:500 dilution in 1% BSA) for one hour at room temperature and analyzed in the CytoFLEX analyzer (Beckman).

### Detection of surface-displayed streptavidin

The expression of surface streptavidin was first detected by western blotting. Secreted streptavidin proteins in the Lp culture were precipitated with 15% trichloroacetic acid (TCA), separated via SDS-PAGE, and transferred onto a 0.22μm nitrocellulose membrane (Bio-Rad). The transferred proteins were visualized on the western blot through detection by a primary c-Myc antibody at a 1:1000 dilution in 3% BSA and a secondary anti-mouse IgG (Cell Signaling #7076) at a 1:1000 dilution in 3% BSA. The membrane-bound streptavidin proteins were dissolved in 8 M urea and visualized using an anti-HA antibody (Thermo Fisher 26183) at a 1:1000 dilution in 3% BSA and the same secondary anti-mouse IgG antibody.

After the confirmation of protein expression, engineered Lp strains were incubated with c-Myc monoclonal or HA tag antibody (1:500 dilution in 1% BSA) for one hour, followed by a one-hour incubation with the secondary anti-mouse antibody (1:500 dilution in 1% BSA) at 25 °C. The antibody-treated cells were then visualized by a Leica DM4000 B fluorescence microscope or analyzed in the CytoFLEX analyzer.

### Loading and release of prodrugs on Lp-Sav

TL-SN was synthesized by WuXi AppTe on a pilot scale. The prodrugs and atto-565 (20 μM) were diluted in PBS to the desired concentrations and incubated with Lp-Sav strains at 37 °C with gentle agitation. The amount of prodrug loaded was determined by a microplate reader recording fluorescence intensity (365 nm excitation and 445 nm emission).

The release of SN was initiated by incubating $H_2O_2$ or GSH with prodrug-loaded Lp-Sav at 37 °C with gentle agitation. The amount of SN released was determined by a microplate reader recording fluorescence intensity (365 nm excitation and 560 emissions) in the supernatant.

### MTT assay

NPC cells were seeded in 96-well plates at a density of 5000 to 8000 per well and cultured for 24 h. SN-38, prodrugs, and prodrug-loaded Lp-Sav strains were added to NPC cells for a 24-h incubation in RPMI 1640 media supplemented with 10% v/v FBS. Both erythromycin and chloramphenicol (100 μg/ml) were added to the coculture to prevent bacterial overgrowth. After 24 h, NPC cells were washed with PBS three times to wash away excessive bacteria after coculture, and MTT (Thermo Fisher) was added to the wells at 0.5 mg/ml. After four hours, the cells and formazan product were dissolved in DMSO, and the viability of the NPC cells was analyzed using a microplate reader. The IC50 was then calculated by SPSS based on the viability of NPC cells.

### Evaluation of the prodrug-loaded Lp-Sav strain in a xenograft NPC mouse model

In this study, three mice per group were used to study the biodistribution of Lp in tumor-bearing mice, and five mice per group were

used to evaluate the performance of prodrug-loaded Lp-Sav strains, totaling 34 male Balb/c nude mice (aged 4–5 weeks; weight 18–22 g; JAX® Mice). Mice are maintained under standard housing conditions (12 light/12 dark cycle, 22–24 °C with humidity set at 40–50%).

Mice were allowed to acclimatize for three days in the animal holding facility before the experiments. Afterward, the mice were subcutaneously injected with $2 \times 10^6$ C666-1 cells in 180 µl of PBS in the right flanks. For the biodistribution study, the tumors were allowed to develop to a tumor volume close to 2000 mm³. Lps expressing the bioluminescence protein CBluc were intravenously injected into tumor-bearing mice through the tail vein at a CFU of $10^8$. Mice were euthanized 24, 48, and 72 h post-injection, and the organs were harvested. To visualize the biodistribution of Lp, the organs were incubated with 2 mg/ml of D-luciferin (GoldenBio) for 10 min and subjected to the IVIS spectrum (PerkinElmer) for in vivo imaging. Next, the organs were homogenized in a FastPrep tissue homogenizer (MP Biomedicals). The homogenized tissues were diluted in PBS and plated on MRS agar plates supplemented with 5 µg/ml erythromycin to calculate the total CFU numbers in various organs.

To evaluate the treatment effectiveness of prodrug-loaded Lp-Sav, PBS, unloaded Lp-Sav strain (Lp-Sav), SN, TL-SN, and Lp-Sav-TL-SN were intravenously administered to tumor-bearing mice following the initial development of subcutaneous C666-1 tumors. Approximately $2 \times 10^8$ CFU of the bacterial strains were administered, and 125 nmol/kg of the drugs were given twice a week, totaling four doses in two weeks. The body weight and tumor volume were recorded for all mice. The tumor-bearing mice were maintained with a maximum allowable tumor volume of 2000 mm³. Mice were euthanized as soon as the tumor volume reached this threshold. The treatment effectiveness was evaluated by comparing the tumor volume and final tumor mass of different groups.

**Reporting summary**

Further information on research design is available in the Nature Portfolio Reporting Summary linked to this article.

## Data availability

All data generated in this study are provided in the Source Data file. Source data are provided with this paper.

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

## Acknowledgements

This work was supported by NUS Medicine Synthetic Biology Translational Research Program (NUHSRO/2020/077/MSC/02/SB, M.W.C.), Investigatorship of the National Research Foundation of Singapore (NRF-NRFI05-2019-0004, M.W.C.), ISF-NRF Joint Program of the National Research Foundation of Singapore (NRF2019-NRF-ISF003-3208, I.Y.H.), the Ministry of Education of Singapore (NUHSRO/2020/046/T1/3, M.W.C.) and the U.S. Air Force Office of Scientific Research – Asian Office of Aerospace Research and Development (FA2386-18-1-4058, M.W.C.). This work used the resources of the Singapore BioFoundry, a bio-manufacturing research facility located at the National University of Singapore. All the schematic diagrams were created with Biorender.com.

## Author contributions

H.S. performed the study and wrote the manuscript. H.S., I.Y.H., Y.S.L, and M.W.C. conceived and designed the study. H.S., C.Z., S.L., Y.L., L.T.L., N.A., K.S.W., J.L., S.P.N., C.W., H.L., J.K.T., D.Y.W., S.Q.Y., I.Y.H., Y.S.L., and M.W.C. analyzed the data. C.Z., Y.L. and S.Q.Y. performed prodrug synthesis and characterization. M.W.C. obtained funding, edited the manuscript, and supervised the study. All authors critically reviewed the manuscript and approved the final version of the manuscript for submission.

## Competing interests

H.S., Y.S.L., I.Y.H. and M.W.C. have filed a patent application through the National University of Singapore based on the work described in this manuscript. The remaining authors declare no competing interests.
