## [Peer Review File · Nature Communications]

REVIEWER COMMENTS

Reviewer #1 (Remarks to the Author):

In this study, Shen et al. developed a bacteria-based prodrug carrier to target nasopharyngeal carcinoma (NPC). The prodrug can be conditionally activated by reactive oxygen species and release toxic chemotherapy compound for anti-tumor therapy. Attempting different strategies to make anticancer drugs more specific and effective is always encouraged, but caution should be considered when using living bacteria for in vivo therapy. Although the data in the manuscript shows that the strategy effectively inhibits the growth of the NPC tumors, the functional mechanism of the drug remains puzzling, which leads to doubts about its effectiveness in treating solid tumors. To improve the study, some concerns need to be addressed.

Major concerns:

- 1, Introduction part, the relationship between commensals and tumor progression needs to be described, as this is the theoretical basis for using bacteria for anticancer drug delivery. Are there any other living bacteria-based drugs approved for clinical trials except for BCG vaccine? Is *L. plantarum* WCFS1 (Lp) a naturally living bacteria in NPC tissues?
- 2, What are the advantages of bacteria-based drugs over antibody-based drugs such as ADC? After all, bacterial particles are much larger and theoretically less conducive to penetrate tissues.
- 3, The current evidence for explaining the mechanism of Lp targeting NPC cells is insufficient, making it difficult to conclude that the interaction of OppA and heparin is the main mechanism of Lp tumor targeting. Firstly, the interaction between these two molecules is not verified, but rather simply inferred by the authors; Secondly, the evidence listed in the article, such as Figure 3e, contradicts this conclusion. This needs to be clarified, as different mechanisms may imply that the bacteria have different tissue affinities.
- 4, Heparin is widely distributed in organs and tissues. If the tumor-targeting ability of Lp was based on the heparin recognition, it may lead to unexpected side effects. Moreover, injecting live bacteria intravenously without testing the blood toxicity is a very irrational practice, that may cause severe sepsis. Why not consider the intra-tumoral injection? The in vivo safety of the bacterial carrier needs to be further explored.
- 5, Figure 4i, why does the EV control strain also bind to TL-SN? Is the difference in binding to SN38 between EV control (4.6 μ M) and Lp-Sav (8.9 μ M) due to the different growth rates of these two strains? Except for streptavidin, are there any other important mechanisms responsible for the loading of prodrugs on the bacterial surface?
- 6, Figure 5, why can TL-SN be activated and release SN38 in cells without the treatment of a redox activator? Does this mean that the prodrug can also be released in advance during their circulation in the body? The explanation for this phenomenon should not be simply "...Lp-Sav strains could be activated by

intrinsic activators from NPC cells". On the other hand, the microenvironment of solid tumors is usually hypoxic, how to ensure that the prodrug can be successfully activated in the tumors?

Minor concerns:

1, line 247, "...with and ..."

2, Line 847, figure 5 legend, c), d), e) and f), g), h), "the activation of TL-SN via H2O2"

Reviewer #2 (Remarks to the Author):

This manuscript describes the successful engineering of a commensal strain to carry and release chemotherapeutic prodrugs in a site-specific manner. The authors produced a manuscript that is well organized and clear in stating their hypotheses, experimental expectations and any precedence which directed their approach. The authors have done substantial work in both the engineering and the in vitro/in vivo experiments that followed. The authors discussed various limitations in their experiments and their reasoning for taking the steps which they did in such a way that the manuscript flowed naturally. The text and figures are in great shape and necessary experimental controls are largely in place. Overall, I feel this is an excellent manuscript from an established investigator in synthetic biology and cell-based therapeutics. I therefore am enthusiastic about publication with only minor revisions.

Specific Comments:

1) There were a few occasions where the authors described unexpected results but did not necessarily follow up on why something did not happen as expected/if said result might confound their main conclusions (one example: Figure 3H, addition of the truncated Lp_0018 SBD did not reduce Lp-NPC binding but did demolish binding to Lp when not cultured with other cell lines.)

2) Include healthy nasal cell line (used earlier in the paper) for flow cytometry and coculture experiments with recombinantly expressed OppA to further confirm specificity of binding between Lp and cancer, and not LP and noncancer cells.

3) KO experiments when culturing Lp and NPC cells with/without purified Lp_0018: knockout gene encoding Lp_0018 and see how binding efficacy is impacted when LpDlp_0018 is incubated with NPCs purified Lp_0018. This is not critical but could help further elucidate the mechanism between purified LP_0018, Lp and NPCs.

4) Unsure if/what controls are used for loading/release experiments (figure 4i)

5) The authors mention in the discussion having issues developing an orthotopic xenograft and thus worked with a xenograft instead. I would assume that application and efficacy of this treatment would be influenced by different factors in the nasal site – an orthotopic model would of course be more insightful in this regard but was not possible, so it is not critical for publication.

Reviewer #3 (Remarks to the Author):

In this present work, the authors identified an oral *Lactobacillus plantarum* (LP) strain capable of binding to nasopharyngeal carcinoma (NPC) through the recognition of heparan sulfate. Furthermore, SN-38 prodrug was loaded on this bacterial vector and delivered to tumor sites actively. The authors proved that LP had specific colonization with the NPC tumor and displayed rapid clearance from non-tumoral tissues based on a xenograft NPC model, which significantly enhanced the potency of SN-38 prodrug in treating both NPC cell lines and NPC mice model. As far as I am concerned, in the current version of this manuscript, there are several lapses in experimental design and more work should be done to support the conclusion. Yet, the idea of this work is interesting and is worth exploring in-depth.

1. What are the morphologies of the designed Lp-Sav-TL-SN or Lp-Sav-BL-SN? A TEM, bio-TEM or SEM are suggested to capture their structures.
2. Page 3, line 92, the author listed nasopharyngeal carcinoma (NPC) affected over 133,000 people and caused over 80,000 deaths in 2020 alone. To make the statistics as fresh as possible, please update it using the most recent statistics e.g., 2022.
3. Fig. 6c and Supplementary Fig. 13, the ex vivo images of major organs and tumors were harvested post-injections. However, the in vivo images indicating the dynamic change of the fluorescence signal in tumor sites should also be provided. Meanwhile, both fluorescence intensity should be quantified in average radiance and compared among groups.
4. Fig. 6d, the colony forming unit (CFU) test of Lp-CB, agar plate photographs of CFU in different organs should be provided in supplementary materials.
5. Fig. 6h, the photographs of excised C666-1 tumors should also be included to intuitively show the therapeutic efficacy and the survival of mice should be observed.
6. The systemic toxicity assessment of this treatment strategy is lacking, e.g., the serum biochemistry parameters, H&E staining after treatments should be analyzed.

7. The authors engineered *Lactobacillus plantarum* strain to deliver SN-38 prodrugs via intravenous injection, will these bacterial vectors be cleared by immune system such as macrophage phagocytosis? Their engulfment effect by phagocytic cells should be evaluated.

8. In materials and methods part, some descriptions of the experiment are too brief and difficult to be repeated. More details such as drug amount, time, why use this specific agent, etc. should be explained, referenced and supplemented. Moreover, many typos are found, please check and correct them throughout the manuscript.

REVIEWER COMMENTS

Reviewer #1 (Remarks to the Author):

In this study, Shen et al. developed a bacteria-based prodrug carrier to target nasopharyngeal carcinoma (NPC). The prodrug can be conditionally activated by reactive oxygen species and release toxic chemotherapy compound for anti-tumor therapy. Attempting different strategies to make anticancer drugs more specific and effective is always encouraged, but caution should be considered when using living bacteria for in vivo therapy. Although the data in the manuscript shows that the strategy effectively inhibits the growth of the NPC tumors, the functional mechanism of the drug remains puzzling, which leads to doubts about its effectiveness in treating solid tumors. To improve the study, some concerns need to be addressed.

Major concerns:

1, Introduction part, the relationship between commensals and tumor progression needs to be described, as this is the theoretical basis for using bacteria for anticancer drug delivery. Are there any other living bacteria-based drugs approved for clinical trials except for BCG vaccine? Is *L. plantarum* WCFS1 (Lp) a naturally living bacteria in NPC tissues?

We appreciate the reviewer's comment. As mentioned in the discussion section, BCG is currently the only FDA-approved bacterial therapy for cancer (lines 465-467). The *L. plantarum* WCFS1 (Lp) strain is identified as a human oral isolate (lines 98-100). However, it's important to note that *L. plantarum* species are found in high abundance in both the nasopharynx (lines 115-116) and NPC (lines 417-421). We have revised these sections for enhanced clarity in response to the reviewer's comments.

Introduction lines 98-100: "In brief, we identified and characterized an oral isolate—the *Lactobacillus plantarum* WCFS1 (Lp) strain—that demonstrates specific binding to NPCs through interactions mediated by the OppA protein and heparan sulfate (Fig. 1a, 1b and 1c)."

Results lines 115-116: "The strains were selected based on their prevalence within their higher species abundance in the nasopharynx³⁰ and NPC²¹, as well as their applications in otolaryngology³¹⁻³⁵."

Discussion lines 417-421: "Diverse carcinoma tissues, including NPC, harbor distinct cancer microbiota, with *Lactobacillus* species constituting a significant component within the microbiome of both carcinoma tissues and the corresponding healthy mucosa²¹⁻²⁵. Therefore, in comparison to many opportunistic pathogens that are non-native to the human microbiome, *Lactobacillus* strains like Lp are readily available in the tumor microenvironment and serves as a safer choice for BMCT applications."

Discussion line 466-469: "As of now, Bacillus Calmette-Guerin (BCG) remains the sole FDA-approved bacterial therapy for cancer. BCG is typically administered locally via intravesical instillation for the treatment of non-muscle-invasive bladder cancer⁷⁸."

2. What are the advantages of bacteria-based drugs over antibody-based drugs such as ADC? After all, bacterial particles are much larger and theoretically less conducive to penetrate tissues.

We appreciate the reviewer's insightful comment. In our study, we chose intravenous administration of Lp strains to evaluate their tumor-targeting efficacy. Nonetheless, our future direction involves the localized

administration of bacteria-based therapeutics directly at the tumor site. This approach, similar to the method of delivering BCG in the bladder, involves administering Lp in the nasal cavity via the mucosal layer and is designed to address the challenges associated with systemic ADC delivery. These challenges include diminished drug bioavailability at the tumor site, extended systemic clearance resulting in organ damage, and additional adverse effect due to antibodies. We have detailed the complexities of ADCs in lines 69-78 and the advantages of localized bacterial delivery in lines 468-472. We have revised these sections to enhance clarity in response to the reviewer's comment.

Introduction lines 69-78: “The conjugation of tumor-targeting carriers could substantially improve the treatment specificity, yet the macromolecular nature of the carriers often complicates the pharmacokinetic profiles of the prodrugs in the circulation, affecting their biodistribution, metabolism and clearance. In the case of ADCs, conjugating the antibody carriers to the native drugs substantially increases the size of the molecule, which impairs the penetration and bioavailability of the drugs in the tumors¹⁷. The longer half-life of the carrier antibodies also prolongs the clearance of payload drugs in the body, damaging liver and kidney functions¹⁸. In addition to systemic toxicity caused by chemotherapy, systemic administration of ADCs can introduce toxicity from the antibody components, inducing immune responses and causing severe secondary injuries that lead to nephrotoxicity in patients¹⁹. Several preclinical studies have also revealed similar complications and side effects for various nanomaterial carriers²⁰”

Discussion lines 469-473: “We anticipate that our designer therapeutics can be administered analogously to BCG through the mucosal layer to target NPCs, thereby circumventing systemic exposure to both the carrier bacteria and chemotherapy agents. This approach holds the potential to mitigate the loss of chemotherapy agents due to first-pass metabolism in the liver and reduce systemic chemotherapy toxicity, observed frequently in the case of ADC, offering a promising avenue for further development.”

3, The current evidence for explaining the mechanism of Lp targeting NPC cells is insufficient, making it difficult to conclude that the interaction of OppA and heparin is the main mechanism of Lp tumor targeting. Firstly, the interaction between these two molecules is not verified, but rather simply inferred by the authors; Secondly, the evidence listed in the article, such as Figure 3e, contradicts this conclusion. This needs to be clarified, as different mechanisms may imply that the bacteria have different tissue affinities.

We appreciate the reviewer's suggestions.

1) Firstly, the interaction between these two molecules is not verified, but rather simply inferred by the authors.

The precise mechanisms of OppA-heparin interactions can be confirmed only through protein structure analysis, which we aim to perform in a future study. In the current study, we have provided substantial evidence supporting the inference of OppA-heparin interactions. Firstly, we predicted the three-dimensional structure of all OppA proteins using AlphaFold and performed heparin docking on the simulated protein models using ClusPro. This illustrated the potential OppA-heparin binding strength of the proteins. Using ClusPro, we obtained superimposed structures of OppA-heparin, heparin binding sites, and hydrogen bond interaction sites (Figure 3c, Supplementary Figures 4 to 7, Supplementary Tables 1 and 2).

Additionally, we further validated the interaction between OppA and heparin through the protein purification process. As previously mentioned (lines 181 to 182), the recombinant OppA proteins were purified using a heparin affinity column, which suggests their heparin-binding properties. We have provided Supplementary Figure 9, which shows the purified OppA proteins, and summarized the elution conditions of the proteins in the heparin affinity column in Supplementary Table 3.

Results lines 181 to 182: “The recombinant proteins were purified through a heparin affinity column and displayed different binding affinities toward heparin (Supplementary Fig. 9).”

Supplementary Figure 9. OppA proteins purified through a heparin affinity column. From lane 1 to 8: Lp_0018 SBD, Lp_0018, Lp_0092, Lp_0200 SBD, Lp_0201, Lp_0783, Lp_1261, Lp_3686.

2) Secondly, the evidence listed in the article, such as Figure 3e, contradicts this conclusion. This needs to be clarified, as different mechanisms may imply that the bacteria have different tissue affinities.

As noted in the manuscript (lines 136-140), our initial hypothesis was that the cancer cell surface heparan sulfate acts as one of the binding targets in the Lp-NPC interaction. This hypothesis was based on previous studies reporting interactions between OppA proteins in *Lactobacillus* and heparan sulfate. To validate this, we incubated NPCs with heparin as a competitive inhibitor and observed a significant reduction in Lp-NPC binding upon heparin addition (Figure 2e).

In Figure 3e, we demonstrated that introducing Lp_0018 enhanced Lp-NPC binding. We hypothesize this enhancement is due to the dual binding property of Lp_0018 to both Lp and NPC (Figure 3f). Further supporting this, the addition of heparin salt to the co-culture decreased the binding enhancement effect of Lp_0018. This finding is consistent with the results shown in Figure 2e and supports our hypothesis that heparan sulfate on NPCs may be a potential binding target for Lp.

Additionally, we engineered Lp_0018 to remove the bacteria-binding domain, resulting in Lp_0018 SBD (Figures 3f and 3g). Introducing Lp_0018 SBD into the co-culture significantly reduced Lp-NPC binding (Figure 3h), confirming that OppA proteins play a critical role in Lp-NPC interactions.

Results lines 136-140: “NPCs are known to overexpress heparan sulfate proteoglycans and have exposed heparan sulfate on the cell surface^{36,37}, which has been reported as a binding target for *Lactobacillus* adhesins³⁸. The respiratory epithelium in the nasal cavity does not present apical

heparan sulfate³⁹, and ciliated HNCs that mimic the nasal epithelium⁴⁰ display negligible binding. For these reasons, we hypothesized that heparan sulfate serves as one of the binding targets in the Lp-NPC interaction.”

4, Heparin is widely distributed in organs and tissues. If the tumor-targeting ability of Lp was based on the heparin recognition, it may lead to unexpected side effects. Moreover, injecting live bacteria intravenously without testing the blood toxicity is a very irrational practice, that may cause severe sepsis. Why not consider the intra-tumoral injection? The in vivo safety of the bacterial carrier needs to be further explored.

As highlighted in our response to Comment 2, we envision administering the prodrug-loaded Lp locally, such as in the mucosal layer of the nasal cavity, to restrict bacterial binding to other tissues. Additionally, heparan sulfate, the binding target for Lp, has been previously reported to be absent in the apical layer of the respiratory mucosa, yet elevated on the surface of NPCs. This characteristic makes heparan sulfate a candidate for precise tumor-targeted drug delivery using Lp. We have discussed this in detail in the following lines:

Results lines 136-139: “NPCs are known to overexpress heparan sulfate proteoglycans and have exposed heparan sulfate on the cell surface^{36,37}, which has been reported as a binding target for *Lactobacillus* adhesins³⁸. The respiratory epithelium in the nasal cavity does not present apical heparan sulfate³⁹, and ciliated HNCs that mimic the nasal epithelium⁴⁰ display negligible binding.”

Discussion lines 462-466: “Our Lp strain has demonstrated the remarkable ability to selectively bind to NPC cells by recognizing heparan sulfate while showing minimal background binding to healthy nasal ciliated cells. This observation closely mirrors the situation in the nasal cavity, where NPC cell surfaces exhibit elevated levels of heparan sulfate due to the overexpression of HSPG, in contrast to the respiratory mucosa within the nasal cavity, where such overexpression is absent^{36,37,39}.”

In this study, due to the technical complexities of developing an orthotopic NPC mouse model, we opted to evaluate our strategy using a xenograft mouse model. To examine the tumor-targeting capability of Lp, we administered the Lp strains via intravenous rather than intra-tumoral injection. We confirmed the safety of the Lp strain by monitoring the colony-forming units (CFU) of Lp in various non-tumoral organs. Our findings indicate that the Lp strain is cleared from non-tumoral organs within 48-72 hours following intravenous injection in mice (Figures 6c and d). This clearance rate is superior to those previously reported for other bacterial-mediated cancer therapies, further substantiating the safety of our Lp strain (lines 414-417).

Discussion lines 414-417: "This clearance of Lp in non-tumoral organs occurs at a rate much faster than that of other tumor-seeking bacteria used in BMCT, including *Salmonella typhimurium* strains⁶⁸, *Listeria monocytogenes*⁶⁹, *Pseudomonas aeruginosa*⁷⁰ and *E. coli*^{71,72}.”

In response to the reviewer's recommendations, and to further demonstrate the safety of Lp, we assessed the live bacterial count in the blood of mice after administering an intravenous injection. We observed that Lp levels rapidly decreased post-injection and became undetectable in the blood six hours afterward (Figure 6e). The swift clearance of bacteria from circulation indicates that intravenous administration of Lp is unlikely to induce sepsis. These findings are detailed in lines 352-356.

Figure 6. e) Bacteria density of Lp-CB in blood 1, 3 and 6 hours after injection. (p value 1 hr vs 3 hrs = 0.0261, 1 hr vs 6 hrs = 0.0172, 3 hrs vs 6 hrs = 0.00029).

Results lines 352-356: “The transient presence of Lp-CB in the organs implies circulation of the bacteria in the mouse bloodstream. To assess the risk of sepsis, we determined the duration of Lp-CB presence in the blood by monitoring the colony-forming units (CFU) of Lp-CB at 1, 3, and 6 hours following intravenous injection (Fig. 6e). Notably, the level of Lp-CB rapidly declined in the mouse blood and became undetectable six hours post-injection (Fig. 6e).”

5, Figure 4i, why does the EV control strain also bind to TL-SN? Is the difference in binding to SN38 between EV control (4.6 μM) and Lp-Sav (8.9 μM) due to the different growth rates of these two strains? Except for streptavidin, are there any other important mechanisms responsible for the loading of prodrugs on the bacterial surface?

We appreciate the reviewer's comment. Both the EV and Lp-Sav strains were incubated with TL-SN in PBS, a medium where bacterial growth is not anticipated. Consequently, we posit that growth rates do not influence the differential loading of TL-SN onto the two bacterial strains.

Biotin, or vitamin B7, is crucial for cell growth, and *Lactobacillus* spp. have been reported to possess multiple endogenous biotin-binding proteins on their membrane. This might have led to the nonspecific attachment of both atto-565 (Figure 4f) and TL-SN (Figure 4i) on the EV strain. Nonetheless, this nonspecific attachment to EV was significantly lower than the loading on Lp-Sav, which is attributed to the specific streptavidin-biotin interaction.

To clarify this point, we have revised the results section from lines 255 to 258 as follows:

“Through direct incubation in PBS, we observed nonspecific attachment of atto-565 to the empty vector (EV) strain, likely due to native biotin-binding proteins present in *Lactobacillus*^{52, 53}. Conversely, the Lp-Sav strain demonstrated significantly higher loading of atto-565 compared to the EV strain (Fig. 4f).”

6, Figure 5, why can TL-SN be activated and release SN38 in cells without the treatment of a redox activator? Does this mean that the prodrug can also be released in advance during their circulation in the

body? The explanation for this phenomenon should not be simply "...Lp-Sav strains could be activated by intrinsic activators from NPC cells". On the other hand, the microenvironment of solid tumors is usually hypoxic, how to ensure that the prodrug can be successfully activated in the tumors?

We appreciate the reviewer's comment. The thioketal linker used in TL-SN is a well-established system activated by reactive oxygen species or glutathione (GSH). The tumor microenvironment is known to exhibit high oxidative stress (doi: 10.1021/acs.jmedchem.0c01704; doi: 10.1155/2021/8532940) and elevated GSH levels (doi: 10.3390/biom10101429) compared to healthy tissues. Several previous studies have demonstrated the activation of thioketal linker-based prodrugs by native redox or GSH signals from tumors (doi: 10.1021/acs.bioconjchem.2c00559.; doi.org/10.1002/adfm.202200791; doi.org/10.1016/j.colsurfb.2020.111223).

In our study, Figures 5c to 5h show that Lp-Sav-TL-SN can be activated via both H₂O₂ and GSH. Given that NPC cells are known to have high levels of both oxidative stress and GSH (doi.org/10.1016/j.imbio.2014.09.021), the activation of Lp-Sav-TL-SN in co-culture with C666-1 is expected (Figure 5i). For further clarification, we have revised the related text to specify "intrinsic activators from NPC cells" and included additional supporting references as follows:

Introduction lines 56-58: "Higher selectivity of prodrugs can be achieved by leveraging physiological conditions unique to the tumor microenvironment (TME), such as hypoxia^{4,5}, acidosis^{6,7}, high oxidative stress^{8,9} and elevated glutathione (GSH) level¹⁰."

Results lines 263-265: "These prodrugs were constructed with two established redox stress-responsive linkers: a thioketal linker⁵⁴⁻⁵⁶ (TL-SN) and a boronic ester linker^{57,58} (BL-SN) (Fig. 4g and Supplementary Fig. 15a)."

Results lines 299-302: "Previous reports have indicated that NPC tumors from patients exhibit high levels of oxidative stress and GSH^{59,60}. Our findings suggest that prodrug-loaded Lp-Sav strains can be activated by these intrinsic activators in NPC cells, leading to the release of SN for targeting the cancer."

Minor concerns:

- 1, line 247, "...with and ..."
- 2, Line 847, figure 5 legend, c), d), e) and f), g), h), "the activation of TL-SN via H₂O₂"

We thank the reviewer for the comment. The typos have been corrected.

Reviewer #2 (Remarks to the Author):

This manuscript describes the successful engineering of a commensal strain to carry and release chemotherapeutic prodrugs in a site-specific manner. The authors produced a manuscript that is well organized and clear in stating their hypotheses, experimental expectations and any precedence which directed their approach. The authors have done substantial work in both the engineering and the in vitro/in vivo experiments that followed. The authors discussed various limitations in their experiments and their reasoning for taking the steps which they did in such a way that the manuscript flowed naturally. The text and figures are in great shape and necessary experimental controls are largely in place. Overall, I feel this is an excellent manuscript from an established investigator in synthetic biology and cell-based therapeutics. I therefore am enthusiastic about publication with only minor revisions.

We thank the reviewer for the comments.

Specific Comments:

1) There were a few occasions where the authors described unexpected results but did not necessarily follow up on why something did not happen as expected/if said result might confound their main conclusions (one example: Figure 3H, addition of the truncated Lp_0018 SBD did not reduce Lp-NPC binding but did demolish binding to Lp when not cultured with other cell lines.)

We appreciate the reviewer's comment. In Figure 3h, we added truncated Lp_0018 SBD to the Lp-NPC co-culture and observed a reduction in Lp-NPC binding across four cancer cell lines, with the exception of CNE-1. For improved clarity, we have updated the results section with a detailed description of the findings related to CNE-1 cells.

Lines 225 to 228: “Additionally, in four of the cell lines, the introduction of Lp_0018 SBD led to a decrease in Lp-NPC binding, ranging from a 25% reduction in RPMI2650 cells to a 65% reduction in C666-1 cells (Fig. 3h). The absence of reduction in CNE-1 cells suggests the presence of additional mechanisms in Lp-CNE-1 binding beyond Lp_0018.”

Furthermore, we have updated the results section to address the unexpected loading of atto-565 on the EV strain (Figure 4f) and the observed limited reduction of IC50 for SN in CNE-2 cells (Figure 5l).

Lines 255-258: “Through direct incubation in PBS, we observed nonspecific attachment of atto-565 to the empty vector (EV) strain, likely due to native biotin-binding proteins present in *Lactobacillus*^{52,53}. Conversely, the Lp-Sav strain demonstrated significantly higher loading of atto-565 compared to the EV strain (Fig. 4f).”

Lines 314-315: “The limited reduction observed in CNE-2 may be attributed to the higher resistance of CNE-2 cells to SN.”

2) Include healthy nasal cell line (used earlier in the paper) for flow cytometry and coculture experiments with recombinantly expressed OppA to further confirm specificity of binding between Lp and cancer, and not LP and noncancer cells.

We previously noted (lines 135 to 139) that the apical layer of the nasal cavity lacks heparan sulfate. However, heparan sulfate is known to be expressed in the basal layer of respiratory epithelium (doi.org/10.1371/journal.ppat.1002986). Similar to respiratory epithelium, the healthy nasal cell (HNC) line consists of various cell types, including multiple basal layers of stem cells rich in heparan sulfate and an apical layer of ciliated nasal epithelial cells. The presence of heparan sulfate in the basal layer of HNC is likely to confound results since flow cytometry analysis necessitates cell dissociation, thereby exposing the basal layer to OppA protein binding.

To address this, we incubated HNC and NPC cells with Lp_0018 during the revision and used immunofluorescence staining to visualize the binding between Lp and cancer cells. As depicted in Supplementary Figures 10 and 11, heparan sulfate on the NPC surface is confirmed by staining of heparan sulfate proteoglycans (HSPG) syndecan-1 and syndecan-2. Lp_0018 bound to NPCs was stained with a cMyc tag antibody, and its distribution resembled that of syndecan-1. In contrast, HNC cells,

which do not express these HSPGs, showed minimal Lp_0018 binding. These findings suggest a preferential binding of Lp_0018 to cancer cells over healthy cells.

We have revised the results section to include these findings in lines 187-191 as follows.

“To further substantiate Lp-cancer cell binding, we incubated Lp_0018 with both NPC and HNC cells and visualized the OppA-NPC binding via immunofluorescence (IF) staining. As depicted in Supplementary Figures 10 and 11, all four NPC types express the HSPGs syndecan-1 and syndecan-2, which are absent in HNC cells. Consequently, Lp_0018 was observed binding to NPCs but not to HNC cells, mirroring the distribution of syndecan-1.”

Supplementary Figure 10. IF staining showing the expression of syndecan-1 and the binding of Lp_0018 to NPCs and HNC. Green – syndecan-1, Red – Lp_0018 stained by anti-cMyc antibody. Blue -- mammalian nucleus stained by Hoechst., Scale bar -- 25 μ m.

Supplementary Figure 11. IF staining showing the expression of syndecan-2 and the binding of Lp_0018 to NPCs and HNC. Green – syndecan-1, Red – Lp_0018 stained by anti-cMyc antibody. Blue -- mammalian nucleus stained by Hoechst., Scale bar -- 25 μ m.

3) KO experiments when culturing Lp and NPC cells with/without purified Lp_0018: knockout gene encoding Lp_0018 and see how binding efficacy is impacted when LpDlp_0018 is incubated with NPCs purified Lp_0018. This is not critical but could help further elucidate the mechanism between purified LP_0018, Lp and NPCs.

We appreciate the reviewer's suggestion. During the revision, we attempted to delete Lp_0018 using a previously reported CRISPR-Cas9 system but were unsuccessful, despite employing three different guide RNAs targeting Lp_0018. As a control, we successfully deleted Lp_0092 using the CRISPR-Cas9 system (Supplementary Figure 12). Notably, the transformation of plasmids for Lp_0018 deletion resulted in severe growth inhibition in the resultant Lp transformants. Given that OppA proteins are crucial for bacterial growth, it is likely that the deletion of Lp_0018 is unfeasible.

We have revised the results section at lines 194-198 to include this finding as follows.

“To further verify the role of Lp_0018 in Lp-NPC binding, we attempted its deletion using a CRISPR-Cas9 system previously described for *Lactobacillus*⁴⁸. However, transformation with the editing plasmid resulted in severe growth inhibition, rendering the deletion of Lp_0018 unsuccessful (Supplementary Fig. 12). As OppA proteins are vital for bacteria to acquire oligopeptides⁴¹, the loss of Lp_0018 is likely detrimental to the Lp strain.”

b

Strain name	Guide RNA sequence
Δ0092	CAAGUCACCCAAGUUGACGA
Δ0018 ¹	GGCCUGACUAGCGAUAUCGU
Δ0018 ²	AGUAUUGCUAUUGUAGCCCU
Δ0018 ³	GCCAGUUAACACUAAUCCUA

Supplementary Figure 12. Deletion of Lp_0018 and Lp_0092. a) Successful deletion of Lp_0092 and unsuccessful deletion of Lp_0018. b) Guide RNA sequences used for the deletion of Lp_0018 and Lp_0092. c) Growth curve of Lp strains bearing the plasmid pLC-0092 and pLC-0018 for Lp_0092 and Lp_0018 deletion.

4) Unsure if/what controls are used for loading/release experiments (figure 4i)

We appreciate the reviewer's comment. An empty vector (EV) control was employed for the loading of TL-SN. In the EV group, nonspecific loading of TL-SN (~ 4.6 μM) was observed, likely due to the endogenous biotin-binding proteins in Lp. We previously mentioned this nonspecific loading while discussing the loading of atto-565.

Lines 255-257: “Through direct incubation in PBS, unspecific loading of atto-565 was observed in the empty vector (EV) strain, likely due to the presence of native biotin binding proteins in *Lactobacillus*^{52,53}. In contrast, the Lp-Sav strain was able to show significantly higher loading of atto-565 than the EV strain (Fig. 4f).”

For the release of SN-38, we have conducted further experiments during the revision and included a PBS control. As illustrated in Supplementary Figure 18, both Lp-Sav-TL-SN and Lp-Sav-BL-SN showed minimal background activation in PBS over a 15-hour incubation period. By the end of the experiment, less than 15% of TL-SN and 5% of BL-SN were released from Lp-Sav. We have updated the results section to reflect this observation.

Lines 279-282: “We initially tested the stability of TL-SN and BL-SN-loaded Lp-Sav strains (referred to as Lp-Sav-TL-SN and Lp-Sav-BL-SN, respectively) in PBS without any activators. Both Lp-Sav-TL-SN and Lp-Sav-BL-SN demonstrated minimal activation and subsequent release of SN over a 15-hour incubation period (Supplementary Fig. 18).”

Supplementary Figure 18. Stability of Lp-Sav-TL-SN and Lp-Sav-BL-SN in PBS. a) & d) Activation of TL-SN and BL-SN with the release of SN from TL-SN-loaded and BL-SN-loaded Lp-Sav over time, respectively. b) & e) Dynamics of TL-SN and BL-SN activation, respectively by spectrum scanning over time. c) & f) Concentration of TL-SN and BL-SN in the cell pellet, respectively and SN in the supernatant pre- and post-activation.

5) The authors mention in the discussion having issues developing an orthotopic xenograft and thus worked with a xenograft instead. I would assume that application and efficacy of this treatment would be influenced by different factors in the nasal site – an orthotopic model would of course be more insightful in this regard but was not possible, so it is not critical for publication.

We are grateful for the reviewer's understanding. Developing an orthotopic NPC model in mice is particularly challenging due to the limited space available in the mouse nasopharynx.

Reviewer #3 (Remarks to the Author):

In this present work, the authors identified an oral *Lactobacillus plantarum* (LP) strain capable of binding to nasopharyngeal carcinoma (NPC) through the recognition of heparan sulfate. Furthermore, SN-38 prodrug was loaded on this bacterial vector and delivered to tumor sites actively. The authors proved that LP had specific colonization with the NPC tumor and displayed rapid clearance from non-tumoral tissues based on a xenograft NPC model, which significantly enhanced the potency of SN-38 prodrug in treating both NPC cell lines and NPC mice model. As far as I am concerned, in the current version of this manuscript, there are several lapses in experimental design and more work should be done to support the conclusion. Yet, the idea of this work is interesting and is worth exploring in-depth.

1. What are the morphologies of the designed Lp-Sav-TL-SN or Lp-Sav-BL-SN? A TEM, bio-TEM or SEM are suggested to capture their structures.

We express our gratitude for the reviewer's suggestion. During the revision, we conducted Transmission Electron Microscopy (TEM) on Lp-Sav-TL-SN and Lp-Sav-BL-SN and compared these to the wild-type Lp strain. As depicted in Supplementary Figure 16, we observed no morphological differences between the prodrug-loaded bacterial strains and the wild-type bacteria. This observation has been incorporated into the results section.

Lines 274-275: "Moreover, Transmission Electron Microscopy (TEM) analysis revealed that the incorporation of both prodrugs did not induce any morphological changes in Lp (Supplementary Fig. 16)."

Supplementary Figure 16. TEM images showing the membrane structure of Lp, Lp-Sav-TL-SN and Lp-Sav-BL-SN. Scale bars from left to right, 1.0 μm , 200.0 nm, 50.0 nm.

2. Page 3, line 92, the author listed nasopharyngeal carcinoma (NPC) affected over 133,000 people and caused over 80,000 deaths in 2020 alone. To make the statistics as fresh as possible, please update it using the most recent statistics e.g., 2022.

We appreciate the reviewer's suggestion. The information presented reflects the most recent statistics on NPC. We have also included a projected incidence rate of NPC for 2040 to offer updated insights into NPC statistics in lines 93 to 95 as follows.

"By 2040, it is projected that the global number of NPC cases and deaths will rise to approximately 179,000 and 114,000, respectively²⁷."

3. Fig. 6c and Supplementary Fig. 13, the ex vivo images of major organs and tumors were harvested post-injections. However, the in vivo images indicating the dynamic change of the fluorescence signal in tumor sites should also be provided. Meanwhile, both fluorescence intensity should be quantified in average radiance and compared among groups.

We are grateful for the reviewer's suggestion. The engineered Lp-CB strain emitted relatively low levels of bioluminescence, insufficient to penetrate mouse skin. Consequently, it was not feasible to monitor the dynamic changes of bioluminescence signals in vivo. Therefore, ex-vivo images of the organs post-

injection were provided to capture the location of Lp-CB. This approach was previously detailed in lines 332 to 334 as follows.

“We then euthanized the mice 24, 48, and 72 hours post-injection, followed by organ harvesting for analysis. This was necessary due to the inability of bioluminescence signals generated by Lp-CB to penetrate the mouse skin.”

We have provided the quantification of bioluminescence intensity in the supporting information. As shown in Supplementary Figure 19d, tumors from the mice given Lp-CB generated the highest bioluminescence signal among all harvested organs at each time point, indicating the accumulation of the bacteria at the tumor site and its rapid clearance from the other organs.

Supplementary Figure 19. IVIS analysis of bioluminescent Lp-CB in mouse organs 72 hours post intravenous injection. a), b) & c) IVIS imaging of the mouse organs 24, 48, 72 hours post injection. d) Quantification of bioluminescence from mouse organs 24, 48, 72 hours post injection.

4. Fig. 6d, the colony forming unit (CFU) test of Lp-CB, agar plate photographs of CFU in different organs should be provided in supplementary materials.

We appreciate the reviewer's suggestion. While we understand the potential value of including photographs of agar plates, the CFU values already provide a quantitative measure of bacterial localization in various organs. Including photographs of agar plates may not significantly enhance the interpretation of the data or alter the overall conclusions. We have aimed to maintain clarity and conciseness in the manuscript, and believe that the inclusion of additional supplementary images of the agar plates is not essential in this context.

5. Fig. 6h, the photographs of excised C666-1 tumors should also be included to intuitively show the therapeutic efficacy and the survival of mice should be observed.

We appreciate the reviewer's suggestion. Photographs of tumors from various treatment groups are now included in Supplementary Figure 21 and referenced in the results section (lines 381-383) as follows:

"Based on tumor volume and the tumor-to-body weight ratio, tumors in mice from the Lp-Sav-TL-SN group were 50% smaller compared to the SN group, 66% smaller than the TL-SN group, and 75% smaller than the PBS control group (Fig. 6h, 6i, and Supplementary Fig. 21)."

Supplementary Figure 21. Representative C666-1 tumors excised from various treatment groups.

6. The systemic toxicity assessment of this treatment strategy is lacking, e.g., the serum biochemistry parameters, H&E staining after treatments should be analyzed.

We are grateful for the reviewer's suggestion. To evaluate the safety of our bacterial vector, we have previously shown in Figures 6c and 6d that Lp is cleared from non-tumoral organs within 72 hours after

intravenous injection in mice. We have emphasized these results in both the results and discussion sections.

Results lines 345-353: “Within 24 hours post-injection, both liver and spleen harbored a greater Lp-CB concentration compared to tumors (Fig. 6d). However, the number of Lp-CB colonizing tumors consistently increased over the course of 72 hours, while Lp-CB levels in the liver and spleen experienced predominant clearance within 48 hours (Fig. 6d). Within the 72-hour timeframe, the tumor-to-liver and tumor-to-spleen ratios for Lp-CB escalated, transitioning from being less than 1:6 and 1:11 to surpassing 200:1 and 35:1, respectively (Fig. 6d). Furthermore, Lp-CB strains were also detected in the heart with a density comparable to that in the tumor and significantly lower levels in other organs, 24 hours postinjection (Fig. 6d). However, bacteria in these organs were no longer detectable 48 hours post-injection (Fig. 6d).”

Discussion lines 412-417: “In tumor-bearing mice, intravenous administration of Lp resulted in the accumulation of bacteria specifically within NPC tumors, with Lp being cleared from non-tumoral tissues 48 hours post-injection. This clearance of Lp in non-tumoral organs occurs at a rate much faster than that of other tumor-seeking bacteria used in BMCT, including *Salmonella typhimurium* strains⁶⁸, *Listeria monocytogenes*⁶⁹, *Pseudomonas aeruginosa*⁷⁰ and *E. coli*^{71,72}.”

Following the reviewer’s suggestion, to further verify the safety of Lp, we have monitored the bacterial count in the blood at 1, 3, and 6 hours following intravenous administration in mice. The bacterial count rapidly declined in the mouse blood over time and was undetectable 6 hours post-injection. Based on these findings, we believe that Lp is unlikely to lead to infection or sepsis. This result is shown in Figure 6e and discussed in the results section.

Results lines 353-357: “The transient presence of Lp-CB in the organs implies circulation of the bacteria in the mouse bloodstream. To assess the risk of sepsis, we determined the duration of Lp-CB presence in the blood by monitoring the colony-forming units (CFU) of Lp-CB at 1, 3, and 6 hours following intravenous injection (Fig. 6e). Notably, the level of Lp-CB rapidly declined in the mouse blood and became undetectable six hours post-injection (Fig. 6e).”

Figure 6. e) Bacteria density of Lp-CB in blood 1, 3 and 6 hours after injection. (p value 1 hr vs 3 hrs = 0.0261, 1 hr vs 6 hrs = 0.0172, 3 hrs vs 6 hrs = 0.00029).

As for SN-38, its pharmacokinetics and systemic toxicity have been extensively studied in previous research. Moreover, we employed a de-escalated regimen to treat NPCs to minimize potential adverse side effects from SN-38. This approach is detailed in the results section.

Results lines 368-374: “We hypothesized that with our strategy, a lower dosage of the prodrug SN would be required to achieve significant inhibition of tumor growth. Therefore, a de-escalating treatment regimen was designed for the animal experiment: the treatments were administered at low doses at 50 µg/kg SN, given twice per week for two weeks, totaling 200 µg/kg SN (Fig. 6a). In comparison, previous studies reported the use of Irinotecan, an FDA approved SN prodrug, at higher total doses ranging from 150 mg/kg to 400 mg/kg in less than four weeks⁶³⁻⁶⁶.”

Additionally, as indicated in Figure 6g, no weight loss was observed in any of the treatment groups throughout the treatment period, suggesting that acute systemic toxicity from the treatment is unlikely.

We wish to emphasize that our study is primarily a proof-of-concept investigation focusing on the novel delivery of payload prodrugs using engineered bacteria. The main objective of our research is to demonstrate the efficacy and targeted delivery capabilities of our strategy. We plan to conduct a thorough investigation of the safety of our proposed treatment in future studies.

7. The authors engineered *Lactobacillus plantarum* strain to deliver SN-38 prodrugs via intravenous injection, will these bacterial vectors be cleared by immune system such as macrophage phagocytosis? Their engulfment effect by phagocytic cells should be evaluated.

We are grateful for the reviewer's suggestion. Prior research has convincingly demonstrated that excessive bacteria are cleared by neutrophilic cells both in the circulatory system and nasal cavity (doi.org/10.4049/jimmunol.157.6.2514; [doi: 10.21053/ceo.2019.00654](https://doi.org/10.21053/ceo.2019.00654)). In the course of our revision, we noted that Lp was indeed rapidly eliminated from circulation, a process likely facilitated by neutrophilic cells, consistent with previous reports.

The primary objective of this proof-of-concept study is to establish the viability and effectiveness of Lp as a drug delivery vector for NPC. Consequently, a detailed evaluation of phagocytic cell engulfment is beyond the immediate purview of this investigation. However, we plan to delve into these aspects in subsequent research endeavors.

8. In materials and methods part, some descriptions of the experiment are too brief and difficult to be repeated. More details such as drug amount, time, why use this specific agent, etc. should be explained, referenced and supplemented. Moreover, many typos are found, please check and correct them throughout the manuscript.

We appreciate the reviewer's suggestion and have revised the methods section to provide more detailed descriptions of the experimental protocols and to clearly specify the source and usage of all reagents.

Lines 518 to 528: "NPC cells were stained with Alexa Fluor 488 phalloidin (Abcam, ab176753) at a 1:1000 dilution in 1% BSA to visualize the cell cytoskeleton. For RPMI 2650 cells, the cytoplasm was highlighted through immunofluorescence (IF) staining of GAPDH using a primary anti-GAPDH mouse monoclonal antibody (Cell Signaling, CST #97166) at a 1:500 dilution in

1% BSA. For HNC cells, micro-cilia were stained through IF staining against acetylated α -tubulin using a primary acetylated α -tubulin rabbit antibody (Abcam, ab209348). Cell-bound Lp_0018 was detected using a primary anti-cMyc antibody (Thermo Fisher MA1-980) at a 1:500 dilution in 1% BSA. Syndecan-1 and syndecan-2 were identified with a primary anti-syndecan-1 antibody (Abcam ab128936) at a 1:200 dilution in 1% BSA and a primary anti-syndecan-2 antibody (Abcam ab205884) at a 1:200 dilution in 1% BSA, respectively. Appropriate secondary antibodies were used to visualize these structures (CST 8889S, 4412S, 4408S, 8890S) at a 1:500 dilution in 1% BSA."

Lines 534 to 535: "For the functional analysis of OppA proteins, Lp_0018, Lp_0018 SBD, and Lp_0200 were added to the Lp-NPC co-culture at a final concentration of 10 μ g/ml."

Lines 551 to 557: "Secreted streptavidin proteins in the Lp culture were precipitated with 15% trichloroacetic acid (TCA), separated via SDS-PAGE, and transferred onto a 0.22 μ m nitrocellulose membrane (Bio-Rad). The transferred proteins were visualized on the western blot through detection by a primary c-Myc antibody at a 1:1000 dilution in 3% BSA and a secondary anti-mouse IgG (Cell Signaling #7076) at a 1:1000 dilution in 3% BSA. The membrane-bound streptavidin proteins were dissolved in 8 M urea and visualized using an anti-HA antibody (Thermo Fisher 26183) at a 1:1000 dilution in 3% BSA and the same secondary anti-mouse IgG antibody."

Supplementary Table 3: Sequence and purification conditions of OppA proteins

Lane	OppA Proteins	Uniprot Entry	Sequence	Purification Condition	Elution Condition
1	Lp_0018 SBD	F9US48	AA 70 to 538	LB media, 100 μ M IPTG, 18°C incubation for 24 hours	1 M NaCl
2	Lp_0018	F9US48	AA 27 to 538	LB media, 100 μ M IPTG, 25°C incubation for 24 hours	1 M NaCl
3	Lp_0092	F9USS1	AA 29 to 519	TB auto-induction media, 30°C incubation for 24 hours	0.5 M NaCl
4	Lp_0200	F9UT07	AA 84 to 553	LB media, 100 μ M IPTG, 25°C incubation for 24 hours	0.35 M NaCl (Further purification through nickel column)
5	Lp_0201	F9UT08	AA 39 to 553	LB media, 100 μ M IPTG, 18°C incubation for 24 hours	0.9 M NaCl

6	Lp_0783	F9UM05	AA 21 to 555	LB media, 100μM IPTG, 25°C incubation for 24 hours	0.7 M NaCl
7	Lp_1261	F9UN51	AA 31 to 547	LB media, 100μM IPTG, 25°C incubation for 24 hours	0.8 M NaCl
8	Lp_3686	F9ULM7	AA 36 to 545	LB media, 100μM IPTG, 25°C incubation for 24 hours	0.8 M NaCl

REVIEWERS' COMMENTS

Reviewer #1 (Remarks to the Author):

In this study, Shen et al. engineered an oral commensal *Lactobacillus plantarum* to construct a tumor-targeting prodrug delivery vector. Their data show that the modified bacteria can deliver the prodrug into the solid tumor and successfully release the chemotherapeutic drug SN-38, thereby inhibiting the growth of tumors such as nasopharyngeal cancer. The revised manuscript effectively addresses the reviewers' concerns, and the supplementary data make the conclusion of the study more solid. Overall, the revised manuscript meets the publication criteria, and it is anticipated that the authors will further optimize the system to explore its practical application under more standardized drug development conditions.

Reviewer #2 (Remarks to the Author):

The authors have done a good job responding to reviewer comments. The manuscript is now ready for publication.

Reviewer #4 (Remarks to the Author):

The manuscript by Shen et al presents the development of bacteria for tumor targeting and specifically for the delivery of prodrugs to the tumor microenvironment. I was invited to step in the stead of Reviewer 3. I have read the manuscript, the reviewer comments, and the response to the comments with interest. Based on my reading, I cannot recommend that this manuscript is accepted to print in Nature Communications.

The predominant reason for my conclusion is that this journal is a premier publication that sets the scene, the standards of practice, and I do not feel that injecting bacteria that are surface-decorated with prodrugs is a viable biomedical strategy. I acknowledge that the manuscript is a very complete story, that the experiments are many and diverse, and the anticancer effects are significant. But the past decades have documented that numerous formulations, too many to count, reveal anticancer activity in vivo. It is time that we start to compare anticancer effects against the best available treatment, not against saline.

I believe the manuscript introduction does not do justice to the field of prodrugs, and specifically the aspect of specificity of drug delivery. This manuscript celebrates the 10-fold change in the IC50 for the prodrugs over the corresponding drug, but in reality this is a very low fold-change. For example, glucuronide prodrugs can afford 100-fold change. ADCs provide a fold-change that is even higher. I can second the opinion of Reviewer 1 that I simply do not see a benefit over the approved ADCs.

Another argument here is the drug loading : relative to a simple prodrug, even the SA_v conjugate is already bulky, then you add a bacteria to carry it... the deliverable payload is simply too low for practical use.

I also second the opinion of the reviewers that injecting bacteria is risky and possibly irresponsible. What is the point ? There doesn't appear to be added benefit from the fact that these are live bacteria. If the cancer homing is simply due to heparin binding, why not take nanoparticles that bind heparin. (and if the response goes that bacteria do proliferate – I would be very skeptical that it would ever be considered in clinic)

All things considered, I believe that this therapeutic approach is un-practical. The manuscript does not offer any benefits over existing treatments.

REVIEWER COMMENTS

Reviewer #1 (Remarks to the Author):

In this study, Shen et al. engineered an oral commensal *Lactobacillus plantarum* to construct a tumor-targeting prodrug delivery vector. Their data show that the modified bacteria can deliver the prodrug into the solid tumor and successfully release the chemotherapeutic drug SN-38, thereby inhibiting the growth of tumors such as nasopharyngeal cancer. The revised manuscript effectively addresses the reviewers' concerns, and the supplementary data make the conclusion of the study more solid. Overall, the revised manuscript meets the publication criteria, and it is anticipated that the authors will further optimize the system to explore its practical application under more standardized drug development conditions.

We thank the reviewer again for the comments.

Reviewer #2 (Remarks to the Author):

The authors have done a good job responding to reviewer comments. The manuscript is now ready for publication.

We thank the reviewer again for the comments.

Reviewer #4 (Remarks to the Author):

The manuscript by Shen et al presents the development of bacteria for tumor targeting and specifically for the delivery of prodrugs to the tumor microenvironment. I was invited to step in the stead of Reviewer 3. I have read the manuscript, the reviewer comments, and the response to the comments with interest. Based on my reading, I cannot recommend that this manuscript is accepted to print in Nature Communications.

The predominant reason for my conclusion is that this journal is a premier publication that sets the scene, the standards of practice, and I do not feel that injecting bacteria that are surface-decorated with prodrugs is a viable biomedical strategy. I acknowledge that the manuscript is a very complete story, that the experiments are many and diverse, and the anticancer effects are significant. But the past decades have documented that numerous formulations, too many to count, reveal anticancer activity in vivo. It is time that we start to compare anticancer effects against the best available treatment, not against saline.

I believe the manuscript introduction does not do justice to the field of prodrugs, and specifically the aspect of specificity of drug delivery. This manuscript celebrates the 10-fold change in the IC50 for the prodrugs over the corresponding drug, but in reality this is a very low fold-change. For example, glucuronide prodrugs can afford 100-fold change. ADCs provide a fold-change that is even higher. I can second the opinion of Reviewer 1 that I simply do not see a benefit over the approved ADCs.

Another argument here is the drug loading : relative to a simple prodrug, even the SA_v conjugate is already bulky, then you add a bacteria to carry it... the deliverable payload is simply too low for practical use.

I also second the opinion of the reviewers that injecting bacteria is risky and possibly irresponsible. What is the point ? There doesn't appear to be added benefit from the fact that these are live bacteria. If the cancer homing is simply due to heparin binding, why not take nanoparticles that bind heparin. (and if the response goes that bacteria do proliferate – I would be very skeptical that it would ever be considered in clinic)

All things considered, I believe that this therapeutic approach is un-practical. The manuscript does not offer any benefits over existing treatments.

We appreciate the reviewer's comment. In the discussion section, we have clarified that our experimental approach involved utilizing a mouse xenograft model of nasopharyngeal carcinoma (NPC). This choice was necessitated by the inherent challenges in developing an orthotopic model for NPC. Specifically, we employed intravenous administration of the engineered commensal bacteria, recognizing this method as a provisional step towards more clinically relevant delivery methods. We anticipate that, in a clinical context, the most effective strategy for administering these bacteria will involve localized mucosal routes. This approach is particularly pertinent to the nasopharynx, where heparan sulfate—a promising target for bacterial adherence and subsequent drug delivery—is abundantly present on the surface of cancer cells but notably absent in healthy respiratory mucosa (lines 463-468 and 473-480).

Furthermore, the introduction section of our manuscript addresses the current limitations in targeting specificity, alongside other challenges associated with existing prodrugs and their delivery systems (lines 69-80). The core aim of our study is to pioneer a platform leveraging commensal bacteria for the precision delivery of prodrugs. Illustrating this, our use of the SN-38 prodrug as a case study revealed a marked suppression in tumor growth, alongside enhanced drug efficacy, attributable to this targeted delivery strategy. We envision that future research will extend our findings by investigating the use of alternative prodrugs possessing lower IC₅₀ values. Such studies will undoubtedly refine our ability to target a broader spectrum of cancers with heightened specificity.

Lastly, in response to the reviewer's feedback, we have thoroughly revised the discussion section to incorporate a comprehensive analysis of the study's limitations (lines 469-474). This amendment ensures a balanced presentation of our findings, facilitating a clearer understanding of the study's context and its implications for future research.

Introduction lines 68-80: Nevertheless, the current prodrug strategies have limited target specificity. For instance, TME cues^{15,16} may not be clearly distinguishable from normal tissue, which would result in significant off-target effects in prodrugs. The conjugation of tumor-targeting carriers could substantially improve the treatment specificity, yet the macromolecular nature of the carriers often complicates the pharmacokinetic profiles of the prodrugs in the circulation, affecting their biodistribution, metabolism and clearance. In the case of ADCs, conjugating the antibody carriers to the native drugs substantially increases the size of the molecule, which impairs the penetration and bioavailability of the drugs in the tumors¹⁷. The longer half-life of the carrier antibodies also prolongs the clearance of payload drugs in the body,

damaging liver and kidney functions¹⁸. In addition to systemic toxicity caused by chemotherapy, systemic administration of ADCs can introduce toxicity from the antibody components, inducing immune responses and causing severe secondary injuries that lead to nephrotoxicity in patients¹⁹. Several preclinical studies have also revealed similar complications and side effects for various nanomaterial carriers²⁰.

Discussion lines 463-468: Due to the challenges associated with developing an orthotopic NPC model, we conducted tests of our strategy using a xenograft mouse model. Our Lp strain has demonstrated the remarkable ability to selectively bind to NPC cells by recognizing heparan sulfate while showing minimal background binding to healthy nasal ciliated cells. This observation closely mirrors the situation in the nasal cavity, where NPC cell surfaces exhibit elevated levels of heparan sulfate due to the overexpression of HSPG, in contrast to the respiratory mucosa within the nasal cavity, where such overexpression is absent^{36,37,39}.

Discussion lines 469-474: Despite this targeted delivery, bacterial-mediated cancer therapy presents several limitations. These include a limited payload loading capacity, interference from native bacteria metabolites, and the potential for infection when administered intravenously. These factors restrict the quantity of bacterial vectors permitted in a single treatment, thus complicating the evaluation of treatment plans in the experimental stage and reducing the likelihood of clinical application. As of now, Bacillus Calmette-Guerin (BCG) remains the sole FDA-approved bacterial therapy for cancer.

Discussion lines 473-480: As of now, Bacillus Calmette-Guerin (BCG) remains the sole FDA-approved bacterial therapy for cancer. BCG is typically administered locally via intravesical instillation for the treatment of non-muscle-invasive bladder cancer⁷⁷. We anticipate that our designer therapeutics can be administered in a manner similar to BCG through the mucosal layer to target NPCs, thereby avoiding systemic exposure to both the carrier bacteria and the chemotherapy agents. This approach holds the potential to mitigate the loss of chemotherapy agents due to first-pass metabolism in the liver and reduce systemic chemotherapy toxicity, observed frequently in the case of ADC, offering a promising avenue for further development.